# Anti-diuretic hormone ITP signals via a guanylate cyclase receptor to modulate systemic homeostasis in *Drosophila*

**Jayati Gera[1], Marishia Agard[2†], Hannah Nave[1†], Austin B Baldridge[3†], Farwa Sajadi[2], Leena Thorat[2], Theresa H McKim[4,5], Shu Kondo[6], Dick R Nässel[7], Mitchell H Omar[3,5], Jean-Paul Paluzzi[2], Meet Zandawala[1,3,5]\***

[1]Neurobiology and Genetics, Theodor-Boveri-Institute, Biocenter, Julius-Maximilians-University of Würzburg, Am Hubland, Würzburg, Germany; [2]Department of Biology, York University, Toronto, Canada; [3]Department of Biochemistry and Molecular Biology, University of Nevada, Reno, United States; [4]Department of Biology, University of Nevada, Reno, United States; [5]Integrative Neuroscience Program, University of Nevada, Reno, United States; [6]Department of Biological Science and Technology, Tokyo University of Science, Tokyo, Japan; [7]Department of Zoology, Stockholm University, Stockholm, Sweden

**\*For correspondence:**
mzandawala@unr.edu

[†]These authors contributed equally to this work

## eLife Assessment

The authors used comprehensive approaches to identify Gyc76C as an ITPa receptor in *Drosophila*. They revealed that ITPa acts via Gyc76C in the renal tubules and fat body to modulate osmotic and metabolic homeostasis. The designed experiments, data, and analyses **convincingly** support the main claims. The findings are **important** to help us better understand how ITP signals contribute to systemic homeostasis regulation.

**Abstract** Insects have evolved a variety of neurohormones that enable them to maintain nutrient and osmotic homeostasis. Here, we characterized the ion transport peptide (ITP) signaling system in *Drosophila*. The *Drosophila ITP* gene can generate three different peptide isoforms: ITP amidated (ITPa) and two ITP-like (ITPL1 and ITPL2) isoforms. We comprehensively characterized the expression of all three ITP isoforms in the nervous system and peripheral tissues. Our analyses reveal widespread expression of ITP isoforms. Moreover, we show that ITPa-producing neurons are activated and release ITPa during dehydration. Furthermore, recombinant *Drosophila* ITPa inhibits diuretic peptide-induced renal tubule secretion ex vivo, thus confirming its role as an anti-diuretic hormone. Using a phylogenetic-driven approach, an ex vivo secretion assay and a heterologous mammalian cell-based assay, we identified and functionally characterized Gyc76C, a membrane guanylate cyclase, as a bona fide *Drosophila* ITPa receptor. Extensive anatomical mapping of Gyc76C reveals that it is highly expressed in larval and adult tissues associated with osmoregulation (renal tubules and rectum) and metabolic homeostasis (fat body). Consistent with this expression, knockdown of Gyc76C in renal tubules impacts tolerance to osmotic and ionic stresses, whereas knockdown specifically in the fat body impacts feeding, nutrient homeostasis, and associated behaviors. We also complement receptor knockdown experiments with *ITP* knockdown and ITPa overexpression in ITPa-producing neurons. Lastly, we utilized connectomics and single-cell transcriptomics to identify pathways via which ITP neurons integrate hygrosensory inputs and interact with other homeostatic hormonal pathways. Taken together, our systematic characterization of ITP signaling establishes a tractable system

to decipher how a small set of neurons integrates diverse inputs to orchestrate systemic homeostasis in *Drosophila*.

## Introduction

Metabolic and osmotic homeostasis are under strict control in organisms to ensure fitness and survival, as well as promote growth and reproduction. Homeostasis is achieved by regulatory mechanisms that impart plasticity to behaviors such as foraging, feeding, drinking, defecation, and physiological processes, including digestion, energy storage/mobilization, and diuresis. For any given homeostatic system, deviations from the optimal range are monitored by external and internal sensors. These, in turn, signal to central neuronal circuits where information about the sensory stimuli and internal states is integrated. The associated regulatory output pathways commonly utilize neuropeptides or peptide hormones to orchestrate appropriate behavioral and physiological processes (*Rajan and Perrimon, 2011*; *Sternson, 2013a*; *Jourjine et al., 2016*; *Lin et al., 2019*; *Nässel and Zandawala, 2019*; *Miroschnikow et al., 2020*; *Benevento et al., 2022*; *McKim et al., 2024*). In mammals, hypothalamic peptidergic neuronal systems, in conjunction with peptide hormones released from the pituitary, are critical regulators of feeding, drinking, metabolic and osmotic homeostasis and reproduction (*Sternson et al., 2013b*; *Saper and Lowell, 2014*; *Le Tissier et al., 2017*; *Benevento et al., 2022*). Several peptidergic pathways have also been delineated in insects that regulate similar homeostatic functions (*Rajan and Perrimon, 2011*; *Schooley et al., 2012*; *Schoofs et al., 2017*; *Lin et al., 2019*; *Nässel and Zandawala, 2019*; *Nässel and Zandawala, 2020*; *Kim et al., 2021*; *Koyama et al., 2023*). Some of these insect pathways originate in the neurosecretory centers of the brain and the ventral nerve cord (VNC), as well as in other endocrine cells located in the intestine (*Raabe, 1989*; *Hartenstein, 2006*; *Zandawala et al., 2018a*; *Nässel and Zandawala, 2020*; *Zandawala et al., 2021*; *Koyama et al., 2023*; *McKim et al., 2024*). Additionally, peptidergic interneurons distributed across the brain also play important roles in regulation of homeostatic behavior and physiology (*Schlegel et al., 2016*; *Martelli et al., 2017*; *Lin et al., 2019*; *Yurgel et al., 2019*; *Miroschnikow et al., 2020*; *Nässel and Zandawala, 2020*). Importantly, some insect neuropeptides are released by both interneurons and neurosecretory cells (NSC), indicating central and hormonal roles, respectively. One such example is the multifunctional ion transport peptide (ITP).

ITP derived its name from its first determined function in the locust *Schistocerca gregaria*, where it increases chloride transport across the ileum and acts as an anti-diuretic hormone (*Audsley et al., 1992b*). Subsequent studies in other insects identified additional roles of ITP, including in reproduction, development, and post-ecdysis behaviors (*Begum et al., 2009*; *Yu et al., 2016b*). In *Drosophila*, ITP influences feeding, drinking, metabolism, and excretion (*Gáliková et al., 2018*; *Gáliková and Klepsatel, 2022*). Moreover, it has a localized interneuronal role in the *Drosophila* circadian clock system (*Johard et al., 2009*; *Hermann-Luibl et al., 2014*; *Reinhard et al., 2024*). The *Drosophila ITP* gene encodes five transcript variants, which generate three distinct peptide isoforms: one ITP amidated (ITPa) isoform and two ITP-like (ITPL1 and ITPL2) isoforms (*Dircksen et al., 2008*; *Gramates et al., 2022*; *Figure 1A*). ITPa is a C-terminally amidated 73 amino acid neuropeptide, while ITPL1 and ITPL2 are non-amidated and possess an alternate, extended C-terminus. ITPa and ITPL isoforms are also found in other insects, indicating conservation of the *ITP* splicing pattern (*Dai et al., 2007*). Moreover, insect ITP is homologous to crustacean hyperglycemic hormone (CHH) and molt-inhibiting hormone (MIH), which together form a large family of multifunctional neuropeptides (*Meredith et al., 1996*; *Dai et al., 2007*; *Drexler et al., 2007*; *Dircksen et al., 2008*; *Begum et al., 2009*; *Webster et al., 2012*).

ITPa is produced by a small set of interneurons and NSC in the *Drosophila* brain and VNC (*Dircksen et al., 2008*). While the expression of the ITPL peptides has not yet been investigated in *Drosophila*, studies in other insects indicate partial overlap with ITPa-expressing neurons (*Drexler et al., 2007*; *Klöcklerová et al., 2023*). In order to delineate the targets of ITPa and ITPL and determine their modes of action, it is first necessary to identify, functionally characterize, and localize the distribution of their cognate receptors. ITPa and ITPL receptors have been characterized in the silk moth *Bombyx mori* (*Nagai et al., 2014*). Surprisingly, *Bombyx* ITPa and ITPL were found to activate G-protein-coupled receptors (GPCRs) for pyrokinin and tachykinin neuropeptides, respectively (*Nagai et al., 2014*; *Nagai-Okatani et al., 2016*). Recently, ITPL2 was also shown to exert anti-diuretic effects via

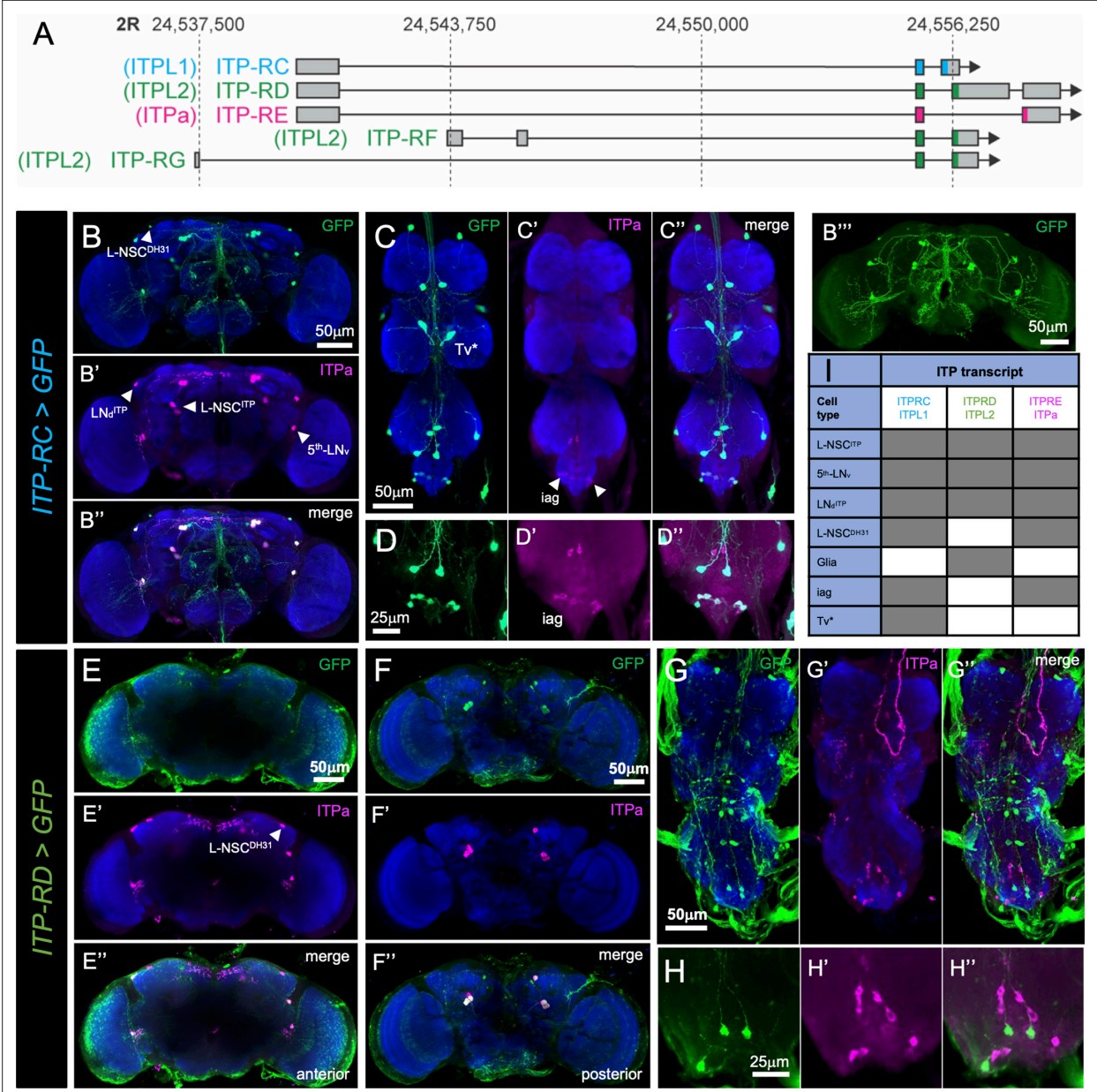

**Figure 1.** *Ion transport peptide (ITP) splicing pattern and expression of ITP transcript variants in the nervous system of adult male* Drosophila. *(**A**)* *Drosophila ITP gene can generate 5 transcript variants (ITP-RC, RD, RE, RF, and RG). ITP-RC encodes ITPL1 precursor, ITP-RD, RF, and RG all encode ITPL2 precursor, and ITP-RE encodes a precursor that produces the amidated ITP (ITPa) peptide. Gray boxes represent exons and lines represent introns (drawn to scale). The regions encoding the open reading frame are colored (pink, green or blue). ITP is located on the second chromosome and numbers on the top indicate the genomic location. ITP-RC-T2A-GAL4 drives GFP (UAS-JFRC81GFP) expression in the (**B**) brain and (**C and D**) ventral nerve cord (VNC). (**B"**) shows another brain preparation (same as in* **Figure 2**) *where axons of ITP-RC neurons are clearly visible. All images are from male flies. Within the brain, ITP-RC is co-expressed with ITPa in four pairs of lateral neurosecretory cells (L-NSC^{ITP}), one pair of diuretic hormone 31 (DH31)-expressing lateral neurosecretory cells (L-NSC^{DH31}), one pair of fifth ventrolateral neurons (5th-LNv) and one pair of dorsolateral neurons (LNd^{ITP}). L-NSC^{ITP} and L-NSC^{DH31} are a subset of lateral neuroendocrine cells and the single pairs of 5th-LNv and LNd^{ITP} belong to the circadian clock network. Within the VNC, ITP-RC is co-expressed with ITPa in abdominal ganglion neurons (iag), which innervate the rectal pad. In addition, ITP-RC is expressed in a pair of Tv\* neurons near the midline in each thoracic neuromere. These neurons are located next to the FMRFamide-expressing Tv neurons (see* **Figure 2**). *ITP-RD-T2A-GAL4 also drives GFP expression in the (**E and F**) brain and (**G and H**) VNC. ITP-RD is expressed in L-NSC^{ITP}, 5th-LNv, and LNd^{ITP} neurons, as well as glia. Within the VNC, ITP-RD is expressed in neurons which are not iag or Tv\* neurons. (**I**) Summary of ITP isoform expression within the nervous system. Gray box indicates presence and white box indicates absence.*

the tachykinin receptor 99D (TkR99D) in a *Drosophila* tumor model (***Xu et al., 2023***). Given the lack of structural similarity between ITPa/ITPL, pyrokinin and tachykinin, the mechanisms governing crosstalk between these diverse signaling pathways are still unclear. More importantly, the presence of any additional ITPa receptors in insects is so far unknown.

Here, we address these knowledge gaps by comprehensively characterizing ITP signaling in *Drosophila*. We used a combination of anatomical mapping and single-cell transcriptome analyses to localize expression of all three ITP isoforms in the nervous system and peripheral tissues. Importantly, we also functionally characterized the membrane-associated receptor guanylate cyclase, Gyc76C, and identified it as a *Drosophila* ITPa receptor. We show that ITPa-Gyc76C signaling to the fat body and renal tubules influences metabolic and osmotic homeostasis, respectively. Lastly, we identified synaptic and paracrine input and output pathways of ITP-expressing neurons using connectomics and single-cell transcriptomics, thus providing a framework to understand how ITP neurons integrate diverse inputs to orchestrate systemic homeostasis in *Drosophila*.

## Results

### Expression of ITP isoforms in the nervous system

In *Drosophila,* the *ITP* gene gives rise to five transcript variants: *ITP-RC*, *-RD*, *-RE*, *-RF,* and *-RG. ITP-RC* encodes an ITPL1 precursor, *ITP-RD, -RF,* and *-RG* all generate an ITPL2 precursor, whereas *ITP-RE* encodes a precursor which yields ITPa (***Figure 1A***). Since the expression of *Drosophila* ITPL isoforms has not yet been mapped within the nervous system, we utilized specific T2A-GAL4 knock-in lines for ITPL1 (*ITP-RC-T2A-GAL4*) and ITPL2 (*ITP-RD-T2A-GAL4*) (***Deng et al., 2019***) to drive GFP expression. Concurrently, we stained these preparations using an antiserum against ITPa (***Hermann-Luibl et al., 2014***) to identify neurons co-expressing ITPa and ITPL isoforms (***Figure 1B–H***).

In agreement with previous reports (***Dircksen et al., 2008***; ***Kahsai et al., 2010***; ***Zandawala et al., 2018b***), ITPa is localized in at least seven bilateral pairs of neurons in the brain (***Figure 1B***). Amongst these are four pairs of lateral NSC, L-NSC$^{ITP}$ also known as ipc-1 (***Dircksen et al., 2008***), or ALKs (***de Haro et al., 2010***), that co-express ITPa, tachykinin, short neuropeptide F (sNPF), and leucokinin (LK) (***Kahsai et al., 2010***; ***Zandawala et al., 2018b***). In addition, there is one pair each of dorsolateral neurons (LN$_d^{ITP}$) and fifth ventrolateral neurons (5$^{th}$-LN$_v$), which are both part of the circadian clock network (***Dircksen et al., 2008***; ***Johard et al., 2009***; ***Reinhard et al., 2024***). Lastly, ITPa is weakly expressed in a pair of lateral NSC (also known as ipc-2a ***Dircksen et al., 2008***). We demonstrate that these neurons (L-NSC$^{DH31}$) co-express diuretic hormone 31 (DH$_{31}$) (***Figure 1B***, ***Figure 2A***). The L-NSC$^{DH31}$ are also referred to as CA-LP neurons since they have axon terminations in the endocrine corpora allata (***Siegmund and Korge, 2001***; ***Kurogi et al., 2023***). Interestingly, all ITPa brain neurons co-express ITPL1 (*ITP-RC*) (***Figure 1B***). L-NSC$^{ITP}$, possibly along with L-NSC$^{DH31}$, are thus a likely source of ITPa and ITPL1 for hormonal release into the circulation.

ITPa expression in the VNC is also sparse and is comprised of only the abdominal ganglion efferent neurons (iag) which innervate the hindgut and rectum (***Figure 1C and D***; ***Dircksen et al., 2008***). In contrast, ITPL1 is expressed more widely, with *ITP-RC-T2A-GAL4* driven GFP detected in iag neurons as well as 14 additional neurons in the VNC (***Figure 1C and D***). Six of these 14 neurons (Tv* in ***Figure 1C***) are located ventrally along the midline and closely resemble the six FMRFamide-expressing Tv neurons in the thoracic ganglia (***Lundquist and Nässel, 1990***; ***O'Brien et al., 1991***). Our analysis reveals that the FMRFamide-expressing Tv neurons are distinct from the ITPL1-expressing ones, although their cell bodies are in close apposition (***Figure 2B***); hence, we refer to these ITPL1-expressing neurons as Tv*. Since abdominal ganglion efferent neurons that produce other neuropeptides have been described previously (***Nässel and Zandawala, 2020***), we asked whether ITPa/ITPL1-expressing iag neurons also express other neuropeptides. Interestingly, iag neurons co-express allatostatin-A (Ast-A) (***Figure 2C, D***) and crustacean cardioactive peptide (CCAP) (***Figure 2E***). In addition, peripheral neurons in the thoracic nerve roots also produce Ast-A and ITPa/ITPL1 (***Figure 2C, D***); however, Ast-A and ITPa/ITPL1 are not co-expressed in the brain (***Figure 2F***).

ITPL2 expression in the brain is also similar to ITPa and ITPL1 (***Figure 1E and F***). However, *ITP-RD-T2A-GAL4* driven GFP was not detected in L-NSC$^{DH31}$ but instead observed in glial cells surrounding the brain. In the VNC, ITPL2 was detected in peripheral glia as well as several neurons not producing ITPa and ITPL1 (***Figure 1G and H***). Taken together, the three ITP isoforms exhibit partial overlapping

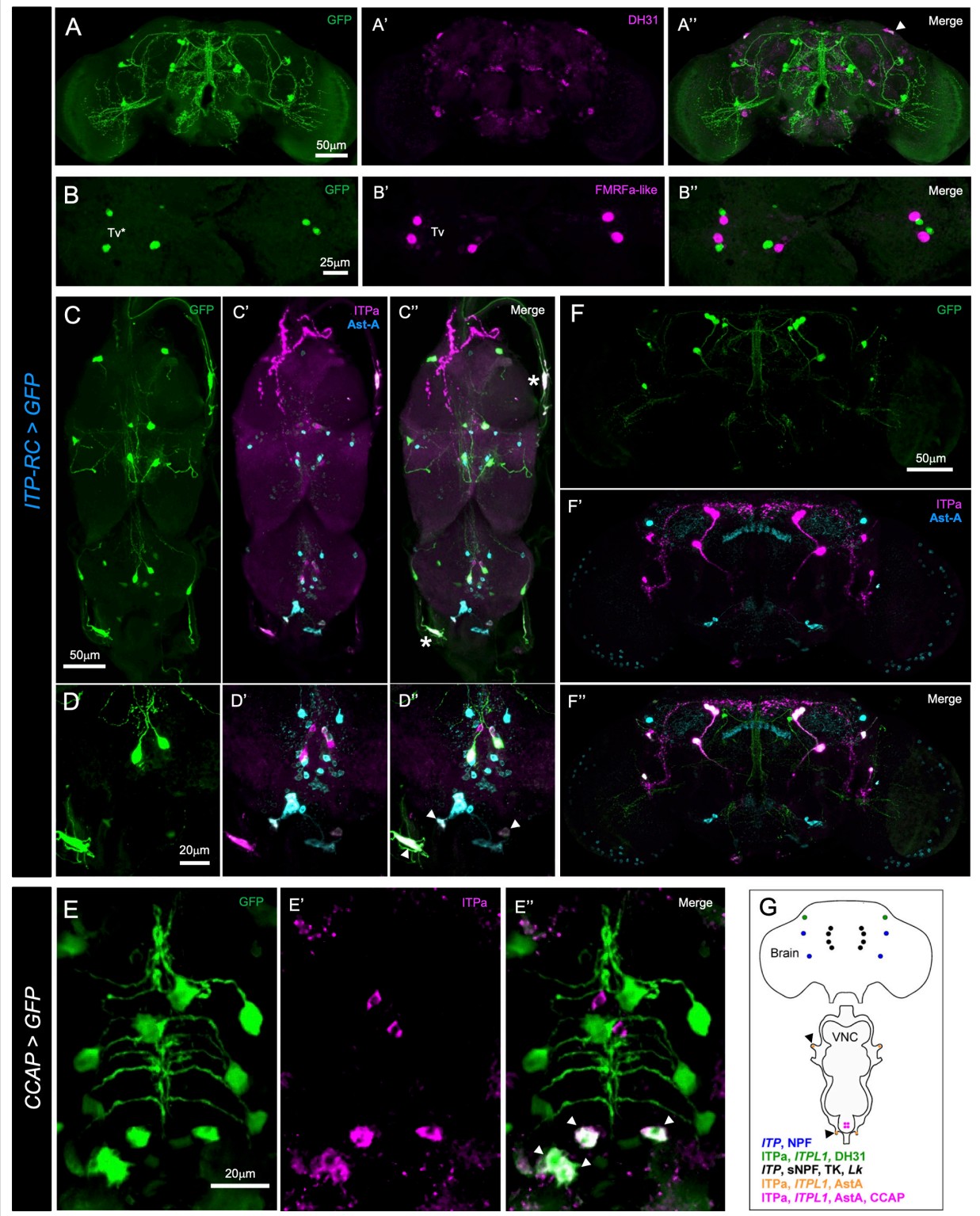

**Figure 2.** *Ion transport peptide (ITP)* is co-expressed with other neuropeptides in the nervous system of adult male *Drosophila*. (**A**) A single pair of *ITP-RC*>GFP-positive lateral neurosecretory cells in the dorsal brain (marked by an arrowhead) co-express diuretic hormone 31 (DH31). (**B**) *ITP-RC* drives GFP expression in a pair of Tv* neurons near the midline in each thoracic neuromere. These neurons are located next to the FMRFamide-expressing Tv neurons. (**C and D**) ITP-RC and ITPa-expressing peripheral neurons (marked by asterisk) and abdominal ganglion neurons (marked by arrowheads) co-express allatostatin-A (Ast-A) neuropeptide. (**E**) *CCAP*>GFP-positive neurons co-express ITPa (and Ast-A by extension) in the abdominal ganglion neurons (marked by arrowheads). (**F**) ITP-RC and ITPa-expressing neurons are distinct from Ast-A-expressing neurons in the brain. (**G**) Schematic of

*Figure 2 continued on next page*

*Figure 2 continued*

the nervous system showing neuropeptides (transcripts or mature peptides) expressed in ITP neurons. Peripheral neurons on one side are marked by arrowheads. Based on previous reports (*Kahsai et al., 2010*; *Hermann-Luibl et al., 2014*; *Zandawala et al., 2018a*) and the present study.

distribution in the nervous system (*Figure 1I*) and are, in some instances, also co-expressed with other neuropeptides in different subsets of neurons (*Figure 2G*).

## Expression of ITP isoforms in peripheral tissues

In the silkworm *Bombyx mori* and the red flour beetle *Tribolium castaneum, ITP* gene products are also expressed outside the nervous system (*Begum et al., 2009*; *Klöcklerová et al., 2023*). This peripheral source of ITP isoforms in *Bombyx* and *Tribolium* includes the gut enteroendocrine cells. In *Bombyx,* peripheral link neurons L1, which innervate the heart also express *ITP*. This prompted us to examine the expression of *Drosophila* ITP isoforms in tissues besides the nervous system. For this, we first examined the global expression of *ITP* using Fly Cell Atlas, a single-nucleus transcriptome atlas of the entire fly (*Li et al., 2022*). Surprisingly, this initial analysis revealed widespread expression of *ITP* across the fly (*Figure 3A–D*). In particular, *ITP* is expressed in the trachea, Malpighian (renal) tubules (MTs), heart, fat body, and gut (*Figure 3B–D*). Fly Cell Atlas only provides expression levels for the entire gene, but not for individual transcript variants. Hence, we next mapped the cellular distribution of individual ITP isoforms in peripheral tissues using the T2A-GAL4 lines and ITPa-immunolabeling. As is the case in *Bombyx*, ITPL1 (*ITP-RC-T2A-GAL4* driven GFP) was detected in peripheral neurons that innervate the heart (*Figure 3E*). In addition, axon terminals of iag neurons which innervate the rectum were also visible (*Figure 3F*). No ITPL1 expression was observed in the fat body, midgut, or MTs (*Figure 3G–I*). Like ITPL1, ITPa immunoreactivity was also detected in a pair of peripheral neurons which innervate the heart and alary muscle (*Figure 3J*), as well as in iag neuron axons that innervate the rectum (*Figure 3K*). ITPa immunoreactivity was not detected in the midgut (*Figure 3L*). In comparison to ITPa and ITPL1, ITPL2 is more broadly expressed in peripheral tissues. Thus, *ITP-RD-T2A-GAL4* drives GFP expression in the heart muscles and the neighboring pericardial nephrocytes (*Figure 3M*), as well as in cells of the middle midgut (*Figure 3N*), posterior midgut (*Figure 3O*), ureter (*Figure 3P*), and trachea (*Figure 3Q*), but not the fat body (*Figure 3R*). In summary, Fly Cell Atlas data are largely in agreement with our comprehensive anatomical mapping of individual ITP isoforms. The widespread expression of *ITP* in peripheral tissues can be largely attributed to ITPL2. Expression of ITPL1 and ITPa, on the other hand, is more restricted and overlaps in cells innervating the heart and rectum.

## Identification of Gyc76C as a putative ITP receptor

ITP has been shown to influence osmotic, ionic, and metabolic homeostasis in insects, including *Drosophila* (*Audsley et al., 1992b*; *Gáliková et al., 2018*; *Gáliková and Klepsatel, 2022*). Considering that the control of hydromineral balance requires stringent integration of all excretory organs, including the rectum and MTs in adult flies, we hypothesized that a putative *Drosophila* ITP receptor would be expressed in these tissues. Our expression mapping of ITP isoforms suggests that osmotic/ionic homeostasis is regulated, at least in part, via a direct effect on the rectum, which is responsible for water and ion reabsorption (*Phillips et al., 1987*; *Coast et al., 2002*; *O'Donnell, 2008*). Additionally, we also expect a putative ITP receptor to be expressed in the fat body, which is a major metabolic tissue. First, we explored if the *Drosophila* orthologs of *Bombyx* ITPa and ITPL receptors (*Figure 4—figure supplement 1*) could also function as ITPa/ITPL receptors in *Drosophila* by examining their expression in the gut, fat body and MTs. *Bombyx* ITPa activates two GPCRs, which are orthologous to *Drosophila* pyrokinin 2 receptor 1 (PK2-R1) and an orphan receptor (CG30340) whose endogenous ligand in *Drosophila* is still unknown (*Nagai et al., 2014*). In addition, *Bombyx* ITPL and tachykinin both activate another GPCR which is related to the *Drosophila* tachykinin receptor at 99D (TkR99D) (*Figure 4—figure supplement 1*; *Nagai et al., 2014*; *Nagai-Okatani et al., 2016*). Our analysis revealed that neither of the three candidate *Drosophila* receptors, PK2-R1, TkR99D, and CG30340, are expressed in the epithelial cells of the rectal pad which mediate ion and water reabsorption (*Figure 4—figure supplement 1*). GFP expression for PK2-R1 and TkR99D was observed in axons innervating the rectum, suggesting that they are expressed in efferent neurons in the abdominal ganglion. In addition, all three receptors were expressed in the midgut or in neurons innervating it (*Figure 4—figure supplement 1*), indicating that the GAL4 drivers and the GFP constructs used here

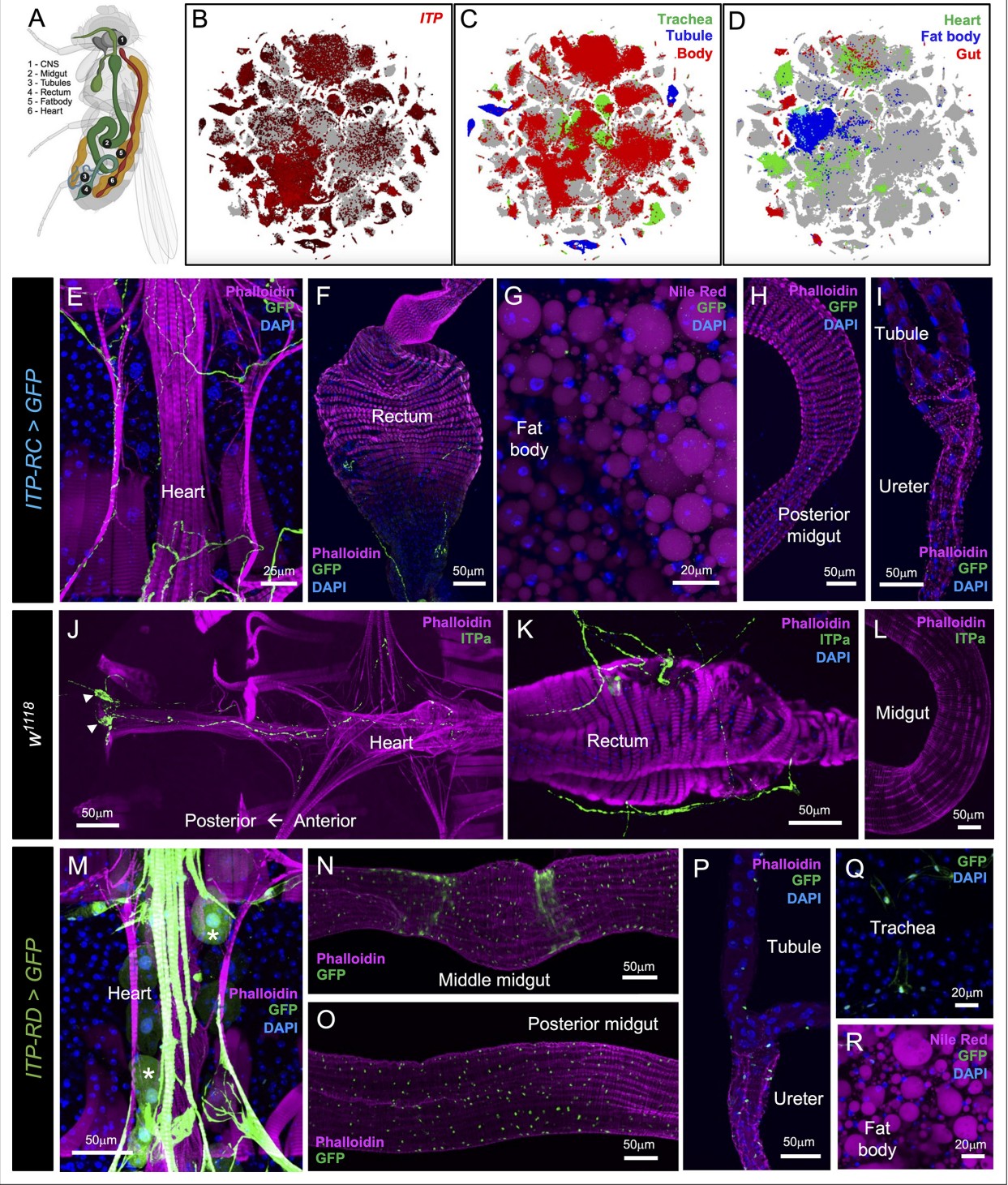

**Figure 3.** *Ion transport peptide (ITP) expression in peripheral tissues of adult male Drosophila.* (**A**) Schematic showing the location of tissues where *ITP* is expressed. Created with BioRender.com. (**B**) t-SNE visualization of single-cell transcriptomes showing *ITP* expression in different tissues of adult *Drosophila*. *ITP* is broadly expressed in peripheral tissues, including (**C**) trachea, Malpighian tubules (tubule), body, (**D**) heart, fat body, and gut. *ITP-RC-T2A-GAL4* drives GFP (UAS-JFRC81GFP) expression in (**E**) peripheral neurons with axons innervating the heart and (**F**) abdominal ganglion neurons which innervate the rectum. ITP-RC is not expressed in (**G**) the fat body, (**H**) midgut, or (**I**) Malpighian tubules. ITPa immunolabeling is present in (**J**) a pair of peripheral neurons (cell bodies marked by arrowheads) innervating the heart and (**K**) in abdominal ganglion neurons which innervate the rectum, but (**L**) absent in the midgut. *ITP-RD-T2A-GAL4* drives GFP expression in (**M**) the heart and nephrocytes (marked by asterisk), (**N**) middle midgut, (**O**) posterior midgut, (**P**) ureter, and (**Q**) trachea. (**R**) ITP-RD is not expressed in the fat body.

are strong enough to report expression in other tissues. Furthermore, single-nucleus RNA sequencing analyses revealed that neither PK2-R1 nor CG30340 are expressed in the cells of the fat body and MTs (*Figure 4—figure supplement 1*). TkR99D, on the other hand, is not detected in the fat body but is expressed in the MT stellate cells (*Figure 4—figure supplement 1*), where it mediates diuretic actions of tachykinin (*Agard et al., 2024*). Hence, expression mapping and/or previous functional analysis of PK2-R1, CG30340, and TkR99D indicates that they are not suited to mediate the anti-diuretic and metabolic effects of ITPa/ITPL. To test this experimentally, we generated recombinant ITPa for analysis of GPCR activation ex vivo. We found that recombinant ITPa failed to activate both TkR99D and PK2-R1 (*Figure 4—figure supplement 1*) heterologously expressed in mammalian CHO-K1 cells. As a control, we showed that their natural respective ligands, tachykinin 1 and pyrokinin 2, resulted in strong receptor activation (*Figure 4—figure supplement 1*). Taken together, these experiments suggest that receptor(s) for *Drosophila* ITPa appear to be evolutionary divergent from *Bombyx* ITPa/ITPL receptors.

Having ruled out the *Bombyx* ITPa/ITPL receptor orthologs as potential candidates, we next employed a phylogenetic-driven approach to identify additional novel ITP receptor(s) in *Drosophila* and other species. Since neuropeptides and their cognate receptors commonly coevolve (*Park et al., 2002*; *Jékely, 2013*), we reasoned that the phyletic distribution of ITP would closely mirror that of a putative ITP receptor. Hence, we first used BLAST and Hidden Markov Model (HMM)-based searches to identify *ITP* genes across all animals. Our analyses retrieved *ITP/CHH/MIH*-like genes in arthropods, nematodes, tardigrades, priapulid worms, and mollusks (*Figure 4A*). A comparison of representative ITP precursor sequences from different phyla reveals that the six cysteines and a few amino acid residues adjacent to them are highly conserved (*Figure 4A*). Thus, *ITP* appears to be restricted to protostomian invertebrates and does not have orthologs in deuterostomian invertebrates and vertebrates. To identify putative orphan receptor(s) which follow a similar phyletic distribution, we performed a phylogenetic analysis of receptors from different vertebrate and invertebrate phyla. We specifically focused on membrane guanylate cyclase receptors (mGC) that all couple with the cGMP pathway because ITP/CHH stimulation has previously been shown to result in an increase in cGMP (*Dircksen, 2009*; *Nagai et al., 2014*). Phylogenetic analysis grouped mGC into six distinct clades (*Figure 4B*). Four of these comprise guanylin, atrial natriuretic peptide (ANP), retinal guanylyl cyclase, and eclosion hormone receptors. Importantly, we retrieved two clades which only contain receptors from protostomian invertebrates (*Figure 4B*). One clade includes *Drosophila* Gyc76C and another includes Gyc32E. Both receptors meet the peptide-receptor co-evolution criteria for ITP receptor identification. However, single-nucleus sequencing data indicate that *Gyc76C* is more highly expressed than *Gyc32E* in MTs (*Figure 4—figure supplement 2*). Independently, we did not detect *Gyc32E-GAL4*-driven GFP expression in MTs and rectal pads, but it was present in the hindgut and fat body (*Figure 4—figure supplement 2*). *Gyc32E-GAL4* is also expressed in a subset of insulin-producing cells (IPCs; labeled with antibody against DILP2) in the brain (*Figure 4—figure supplement 2*). Thus, the lack of *Gyc32E* expression in osmoregulatory tissues, coupled with the fact that Gyc76C was previously implicated in the ITP signaling pathway in *Bombyx* (*Nagai et al., 2014*), prompted us to focus on Gyc76C further.

## Gyc76C expression in Drosophila

If Gyc76C functions as an ITP receptor in *Drosophila*, it should be expressed in cells and tissues that are innervated by ITPa/ITPL-expressing neurons, as well as in tissues, which mediate some of the known hormonal functions of ITP. To validate this prediction, we used a recently generated T2A-GAL4 knock-in line for Gyc76C (*Figure 5A*; *Kondo et al., 2020*) to comprehensively map its expression throughout larval *Drosophila* (*Figure 5—figure supplement 1*) and in adult males (*Figure 5B–O*) and females (*Figure 5—figure supplement 2*). In males, *Gyc76C-T2A-GAL4* drives GFP expression throughout the adult intestinal tract, including the anterior midgut (*Figure 5B*), ureter (of renal tubules) (*Figure 5C*), and posterior midgut (*Figure 5D*). Importantly, in agreement with the role of *Drosophila* ITP in regulating osmotic (*Gáliková et al., 2018*) and metabolic homeostasis (*Gáliková and Klepsatel, 2022*), Gyc76C is highly expressed in the renal tubules (*Figure 5E*), rectum (*Figure 5F*), and adipocytes of the fat body (*Figure 5G*). Moreover, we see a convergence of ITPa-immunolabeled axon terminations and Gyc76C expression in the anterior midgut (*Figure 5H*) and the rectal papillae in the rectum (*Figure 5I*), the latter of which are important for water reabsorption as first proposed nearly

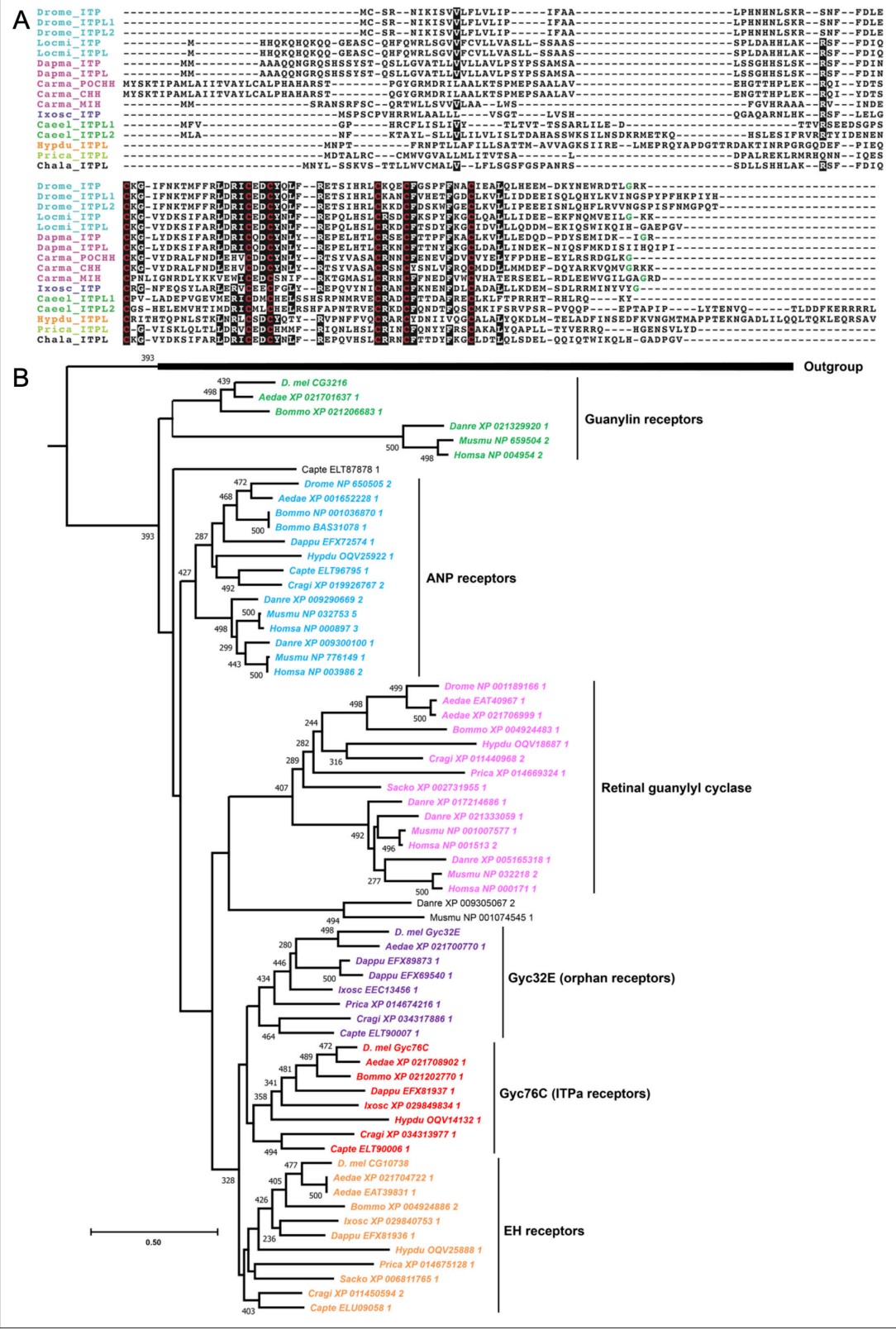

**Figure 4.** Ion transport peptide (ITP) signaling components are found in protostomes. (**A**) Multiple sequence alignment of ITP precursor sequences. ITP is homologous to crustacean hyperglycemic hormone (CHH) and molt-inhibiting hormone (MIH). Note the conservation of six cysteine residues (highlighted in red) across all the species. C-terminal glycine, which is predicted to undergo amidation is colored in green. Species abbreviations: Drome, *Drosophila melanogaster*; Locmi, *Locusta migratoria*; Dapma, *Daphnia magna*; Carma, *Carcinus maenas*; Ixosc, *Ixodes scapularis*; Caeel,

*Figure 4 continued on next page*

Figure 4 continued

*Caenorhabditis elegans*; Hypdu, *Hypsibius dujardini*; Prica, *Priapulus caudatus*; Chala, *Charonia lampas*. (**B**) Maximum-likelihood phylogeny of membrane guanylate cyclase receptors identifies two clades that are restricted to protostome phyla which also have ITP. The clade containing *D. melanogaster* Gyc76C receptor are the putative ITPa receptors. Bootstrap values higher than 200 (based on 500 replicates) are indicated adjacent to the nodes. *Drosophila* guanylate cyclase alpha and beta subunits were used as outgroups.

The online version of this article includes the following figure supplement(s) for figure 4:

**Figure supplement 1.** *Drosophila* orthologs of *Bombyx mori* ion transport peptide (ITP) and ITPL receptors.

**Figure supplement 2.** Expression of candidate ITP amidated (ITPa)/ITPL receptors in adult male *Drosophila* tissues.

a century ago (*Wigglesworth, 1932*). Gyc76C is also broadly expressed in neurons throughout the brain (*Figure 5J*) and VNC (*Figure 5K*). Consistent with the role of ITP in regulating circadian rhythms, Gyc76C is expressed in glia clock cells (*Figure 5L*), and subsets of dorsal clock neurons (labeled with antibody against the clock protein Period) that are near the axon terminations of the clock neurons LNd[ITP] and 5th-LN$_v$ (*Figure 5M*). Gyc76C is not expressed in lateral clock neurons which are situated more closely to ITPa-expressing clock neurons (*Figure 5L*). Similar to males, *Gyc76C-T2A-GAL4* also drives GFP expression in the female fat body, renal tubules, midgut, brain, VNC and subsets of dorsal clock neurons (*Figure 5—figure supplement 2*). Interestingly, Gyc76C is not expressed in male IPCs (labeled with antibody against DILP2) (*Figure 5N*) but is expressed in a subset of female IPCs (*Figure 5—figure supplement 2*). However, Gyc76C is not expressed in endocrine cells producing glucagon-like adipokinetic hormone (AKH) in either males (*Figure 5O*) or females (*Figure 5—figure supplement 2*). Interestingly, L-NSC[DH31], which innervate the corpora allata, might utilize both DH$_{31}$ and ITPa/ITPL1 to modulate juvenile hormone production since Gyc76C is expressed in the corpora allata (*Figure 5O*). Lastly, we also explored the distribution of Gyc76C in larval tissues, where expression was detected in the adipocytes of the fat body, all the regions of the gut and in renal tubules (*Figure 5—figure supplement 1*). Gyc76C is widely distributed in the larval nervous system, with high expression in the endocrine ring gland, where ITPa-immunoreactive axons terminate (*Figure 5—figure supplement 1*). Taken together, the cellular expression of Gyc76C in the nervous system and peripheral tissues of both larval and adult *Drosophila* further indicates that it could mediate the known effects of ITP.

## Gyc76C is necessary for ITPa-mediated inhibition of renal tubule secretion ex vivo

ITP has been shown to modulate osmotic homeostasis in *Drosophila* by suppressing excretion (*Gáliková et al., 2018*). While the precise mechanisms underlying the anti-diuretic effects of *Drosophila* ITP are not known, previous research in other systems provide important insights. For instance, in the locust *Schistocerca gregaria*, ITPa but not ITPL promotes ion and water reabsorption across the hindgut (*Audsley et al., 1992a*; *Audsley et al., 1992b*; *King et al., 1999*; *Wang et al., 2000*), thereby promoting anti-diuresis. Previous experiments have shown that the MTs are also targeted by anti-diuretic hormones: CAPA neuropeptides inhibit diuresis in some insects, including *Drosophila* via direct hormonal actions on the renal tubules (*Paluzzi et al., 2008*; *MacMillan et al., 2018*; *Sajadi et al., 2020*; *Sajadi et al., 2023*). Given the expression of Gyc76C, our candidate ITP receptor, in both the hindgut and renal tubules, ITP could modulate osmotic and/or ionic homeostasis by targeting these two excretory organs. Hence, we utilized the Ramsay assay (*Figure 6A*) to monitor ex vivo fluid secretion by MTs in response to application of recombinant ITPa. Interestingly, recombinant ITPa does not influence rates of secretion by unstimulated tubules (*Figure 6B*). Since the basal secretion rates are quite low, we tested if ITPa can inhibit secretion stimulated by LK, a diuretic hormone targeting stellate cells (*O'Donnell et al., 1996*), and a calcitonin-related peptide, DH$_{31}$, which acts on principal cells (*Johnson et al., 2005*). Recombinant ITPa inhibits LK-stimulated secretion by MTs from $w^{1118}$ flies (*Figure 6C*), indicating that stellate cell-driven diuresis is sensitive to this anti-diuretic hormone. Similarly, DH$_{31}$-stimulated secretion was also inhibited by recombinant ITPa (*Figure 6D*), demonstrating that principal cells are also modulated by ITPa. This result confirmed that the effect of ITPa on osmotic homeostasis are mediated, at least partially, via actions on renal tubules and that both major cell types are targeted.

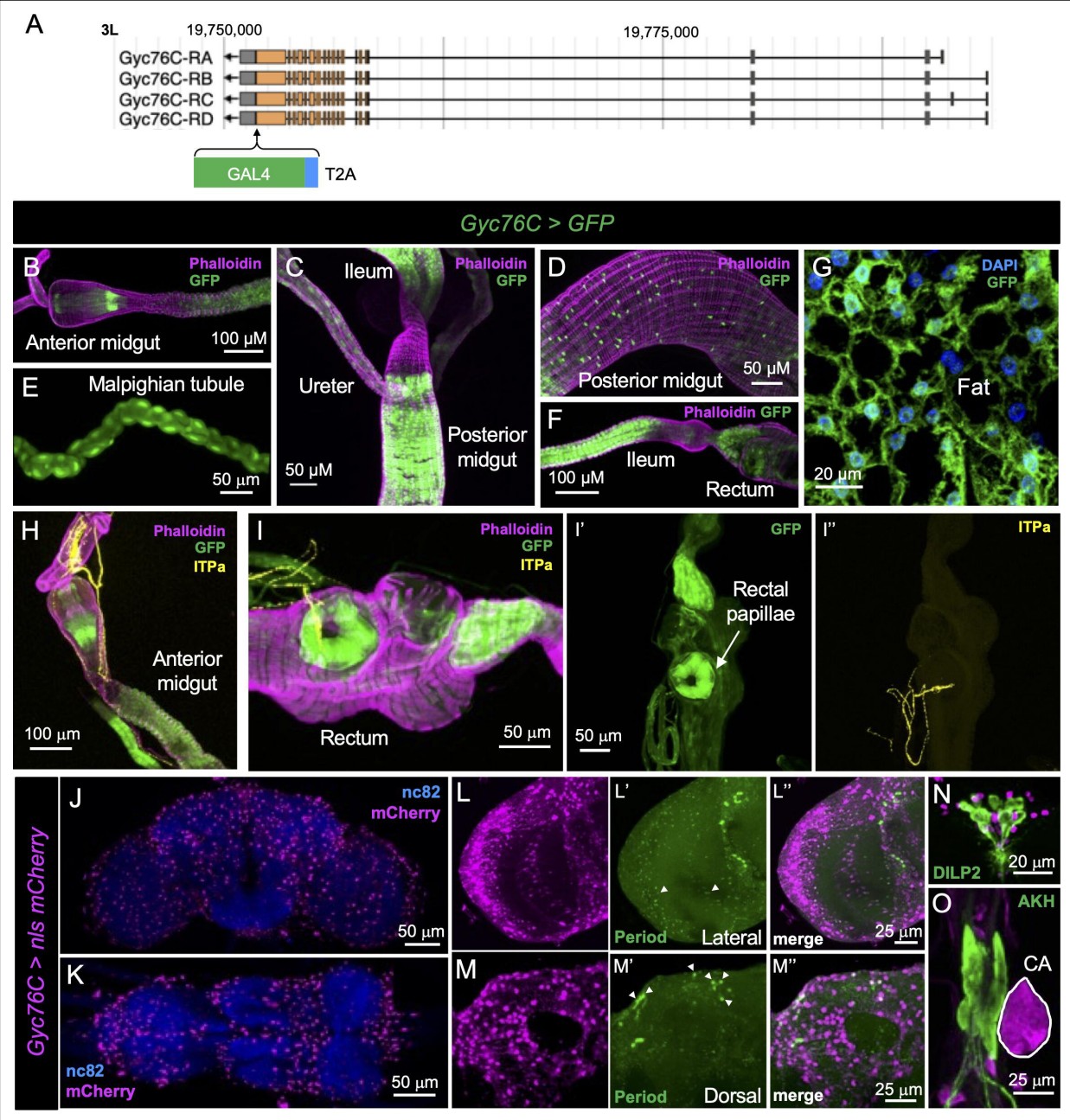

**Figure 5.** Gyc76C expression in adult male *Drosophila*. (**A**) Schematic showing the generation of *Gyc76C-T2A-GAL4* knock-in line. *Gyc76C-T2A-GAL4* drives GFP (UAS-JFRC81GFP) expression in the (**B**) anterior midgut, (**C**) ureter, (**D**) posterior midgut, (**E**) Malpighian tubules, (**F**) ileum, rectum, (**G**) and adipocytes in the fat body. Gyc76C is expressed in the regions of (**H**) the anterior midgut and (**I**) rectal papillae in the rectum that are innervated by ITP amidated (ITPa)-expressing neurons. Gyc76C is also broadly expressed in the (**J**) brain and (**K**) ventral nerve cord. (**L**) Gyc76C is expressed in glial clock cells and (**M**) subsets of dorsal clock neurons (both labeled by Period antibody and marked by arrowheads). Gyc76C is not expressed in (**N**) insulin-producing cells (labeled by DILP2 antibody) and (**O**) adipokinetic hormone (AKH) producing endocrine cells but is expressed in the corpora allata (CA) (marked in white).

The online version of this article includes the following figure supplement(s) for figure 5:

**Figure supplement 1.** Gyc76C expression in larval *Drosophila*.

**Figure supplement 2.** Gyc76c expression in adult female *Drosophila*.

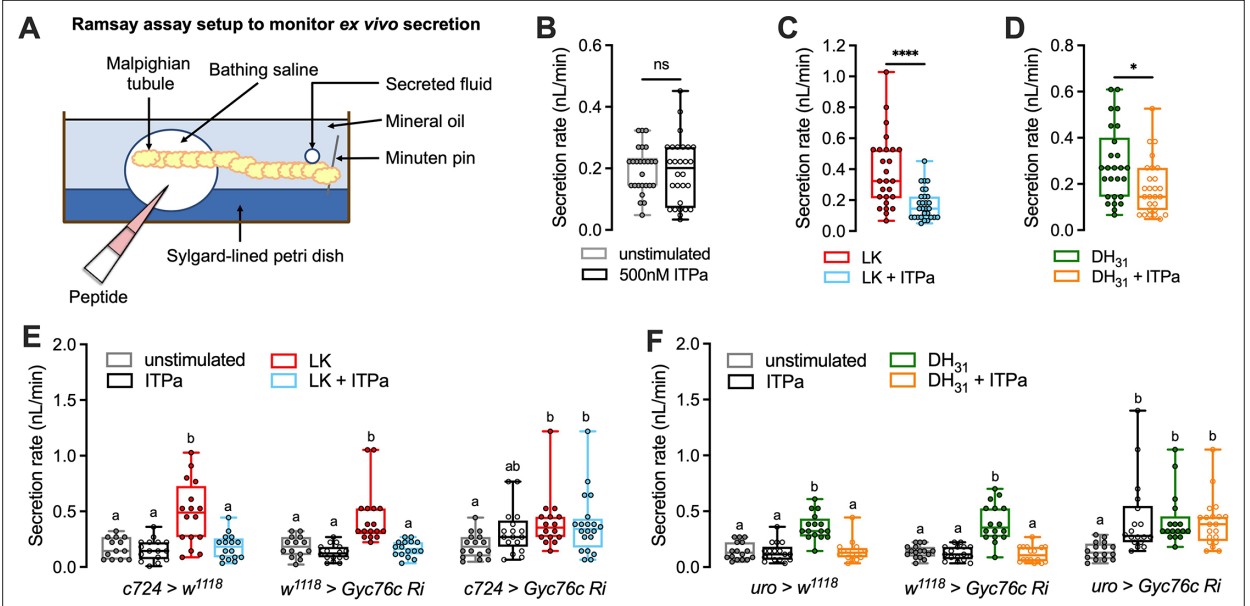

**Figure 6.** Recombinant *Drosophila* ITP amidated (ITPa) inhibits Malpighian tubule secretion via Gyc76C. (**A**) Schematic of Ramsay assay used to monitor ex vivo secretion by tubules. (**B**) Application of *Drosophila* 500 nM ITPa does not affect basal secretion rates by unstimulated tubules. 500 nM ITPa inhibits both (**C**) 10 nM leucokinin (LK)-stimulated and (**D**) 1 µM diuretic hormone 31 (DH$_{31}$)-stimulated secretion rates. Importantly, while 500 nM ITPa inhibits (**E**) 10 nM LK-stimulated secretion and (**F**) 1 µM DH31-stimulated by renal tubules from control flies, this inhibitory effect is abolished in tubules where *Gyc76C* has been knocked down with *UAS-Gyc76C RNAi* (#106525) in stellate cells using the *c724-GAL4* and in principal cells using *uro-GAL4*. Male Malpighian tubules were used for all experiments. For (**B**-**D**) $p<0.05$ and ****$p<0.0001$ as assessed by unpaired *t*-test. For (**E** and **F**), within each genotype, different letters denote secretion rates that are significantly different from one another ($p<0.05$) as assessed by two-way ANOVA followed by Tukey's multiple comparisons test.

The online version of this article includes the following figure supplement(s) for figure 6:

**Figure supplement 1.** Recombinant *Drosophila* ITP amidated (ITPa) inhibits Malpighian tubule secretion via Gyc76C.

**Figure supplement 2.** Western blot analysis of recombinant ITP amidated (ITPa) produced in AtT-20 cells.

We next utilized the Ramsay assay to assess if Gyc76C is necessary for the inhibitory effects of ITPa on renal tubule secretion. Notably, knocking down expression of Gyc76C in stellate cells using *c724-GAL4* abolished the anti-diuretic action of ITPa in LK-stimulated tubules (**Figure 6E**). Similarly, tubules in which *Gyc76C* was knocked down using the *LK receptor GAL4* (**Zandawala et al., 2018b**) do not exhibit reduced secretion following ITPa application (**Figure 6—figure supplement 1**). Additionally, knocking down expression of Gyc76C in principal cells using *uro-GAL4* abolished the anti-diuretic action of ITPa in DH$_{31}$-stimulated tubules (**Figure 6F**). Surprisingly, the application of ITPa alone promotes fluid secretion compared to unstimulated controls in tubules with Gyc76C knockdown (**Figure 6E and F**). This effect is more prominent in tubules with Gyc76C knockdown, specifically in the principal cells (**Figure 6E and F**). This suggests that ITPa could also interact with other yet unknown receptors in addition to Gyc76C. Nonetheless, these results indicate that ITPa exerts its anti-diuretic effects on MTs via Gyc76C, which acts as a functional ITPa receptor in both stellate and principal cells of the tubules.

## ITPa activates Gyc76C in HEK293T cells

To further test whether ITPa activates Gyc76C, we leveraged a heterologous expression assay. HEK293T cells were transfected with Gyc76C-HA and/or the fluorescent protein-based cGMP indicator Green cGull (**Matsuda et al., 2017**) and subjected to live-cell imaging (**Figure 7A**). At 4 min after 50 nM, 250 nM, or 500 nM ITPa treatment, cells expressing HA-tagged Gyc76C and Green cGull exhibited 34%, 88.7%, and 111.4% increases in fluorescence intensity, respectively, indicating a dose-dependent increase in cGMP (**Figure 7B-E**, **Figure 7—figure supplement 1**). The same measurements in control cells transfected with only Green cGull failed to demonstrate a clear dose-dependent response with respective increases of 17.5%, 39.8%, and 28.2% (**Figure 7**, **Figure 7—figure supplement 1**).

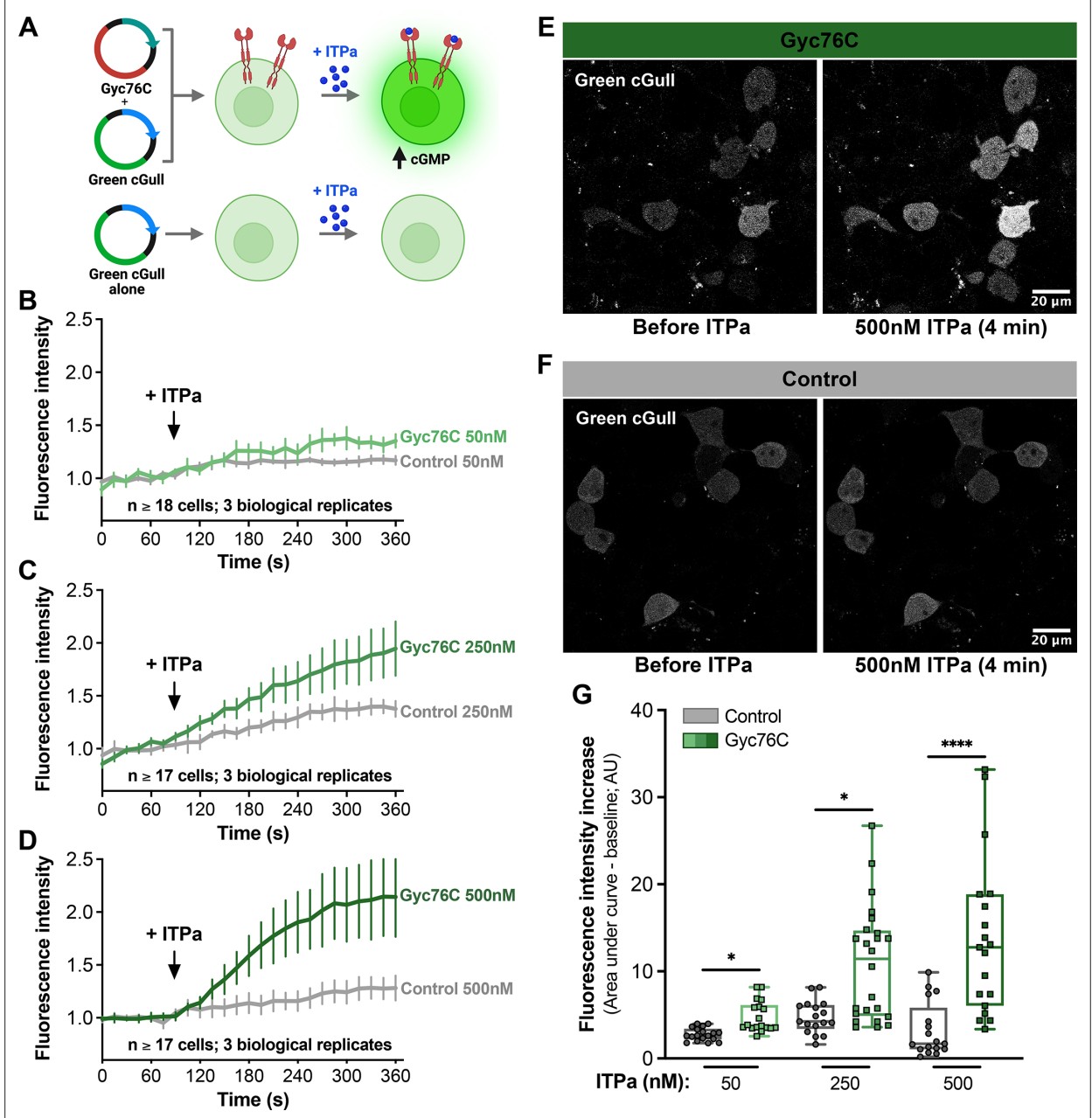

**Figure 7.** ITP amidated (ITPa) activates Gyc76C heterologously expressed in HEK293T cells. (**A**) Schematic of the heterologous assay used to functionally characterize Gyc76C. Created with BioRender.com. Application of (**B**) 50 mM, (**C**) 250 mM, or (**D**) 500 nM *Drosophila* ITPa to HEK293T cells transiently expressing Green cGull (cGMP sensor) and Gyc76C results in a dose-dependent increase in fluorescence compared to control cells which do not express Gyc76C. Graphs represent the mean fluorescence of 17–18 cells. Representative images showing fluorescence in (**E**) HEK293T cells expressing Gyc76C and (**F**) those without Gyc76C before and 4 min after the addition of 500 nm ITPa. (**G**) Area under the curve analysis demonstrates significant differences in Green cGull fluorescence increases between experimental and control conditions; *$p<0.05$ and ****$p<0.0001$ as assessed by nonparametric one-way ANOVA followed by Dunn's test for multiple comparisons.

The online version of this article includes the following figure supplement(s) for figure 7:

**Figure supplement 1.** ITP amidated (ITPa) activates Gyc76C in HEK293T cells.

Area under the curve analysis confirmed significant differences between Gyc76C and control conditions (*Figure 7G*). Exogenous protein expression levels were confirmed with post-hoc staining of Gyc76C-HA and Green cGull (*Figure 7—figure supplement 1*). Several cells were found to express Green cGull independent of Gyc76C expression, explaining absence of strong response to ITPa in

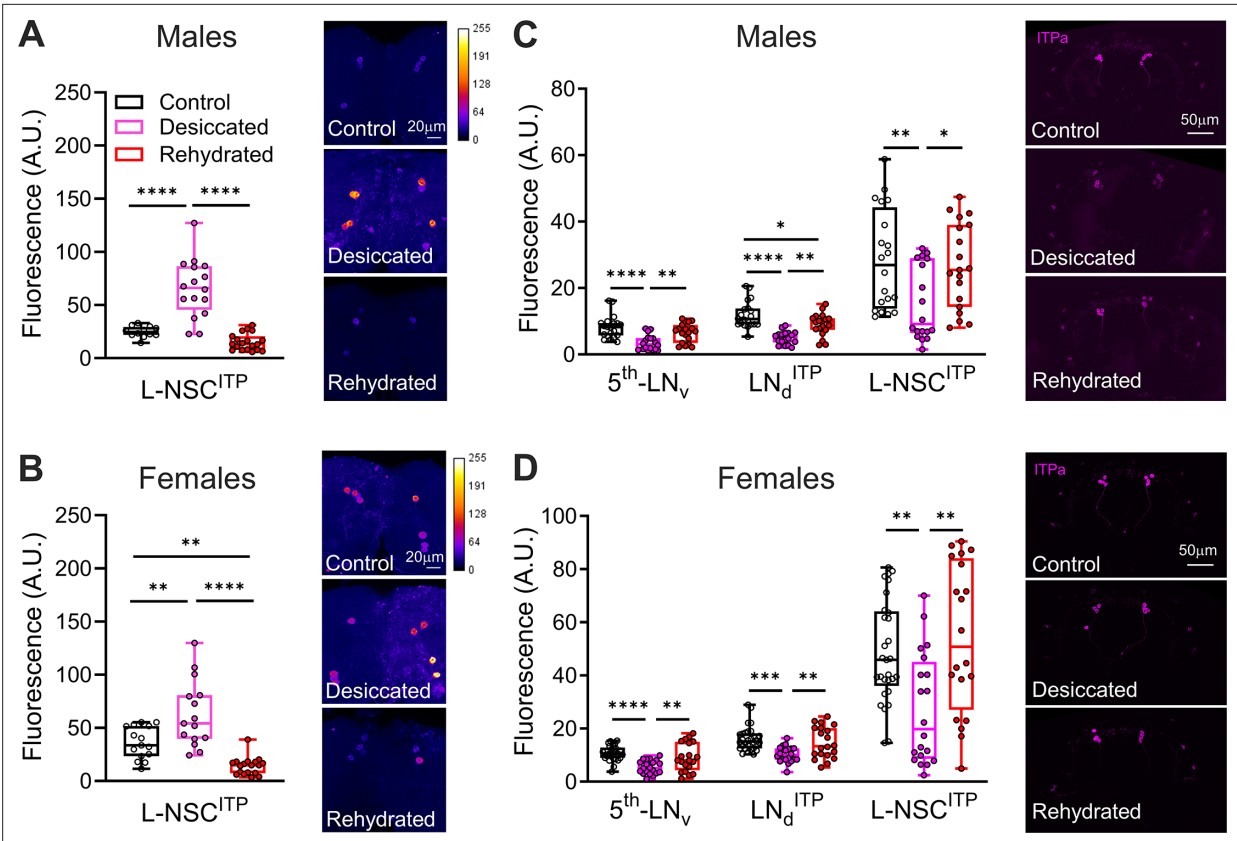

**Figure 8.** ITP amidated (ITPa) neurons are active and release ITPa during desiccation. GFP immunofluorescence, indicative of calcium levels and measured using the CaLexA reporter, is increased in L-NSC[ITP] of (**A**) male and (**B**) female flies exposed to desiccation. The GFP intensity returns to control levels in flies that were rehydrated following desiccation. ITPa immunofluorescence, indicative of peptide levels, is lowered in 5th-LN[v], LN[d][ITP] and L-NSC[ITP] of (**C**) male and (**D**) female flies exposed to desiccation. ITPa peptide levels recover to control levels in flies that were rehydrated following desiccation. Lower peptide levels during desiccation indicate increased release. For all panels, *$p<0.05$, **$p<0.01$, ***$p<0.001$, ****$p<0.0001$ as assessed by one-way ANOVA followed by Tukey's multiple comparisons test.

some cells. Finally, staining intensities were plotted against peak live-cell fluorescence increases, revealing weak negative correlations between exogenous protein expression and increases in cGMP signal (*Figure 7—figure supplement 1*). The observed correlations may reflect lower dynamic ranges due to higher baseline fluorescence in cells receiving more exogenous DNA. Taken together, these live-imaging results strongly suggest that ITPa can induce increased cGMP production through Gyc76C.

## ITPa-producing neurons are activated and release ITPa under desiccation

Having validated Gyc76C as a functional ITPa receptor, we next wanted to determine the context(s) during which ITP signaling is active in vivo. ITP expression has been shown to be upregulated during desiccation (*Gáliková et al., 2018*). However, whether this increased transcription is also coupled with increased ITP signaling is unknown. Consistent with its role as an anti-diuretic hormone, we hypothesized that ITP signaling is increased under desiccation. To test this hypothesis, we employed CaLexA (*Masuyama et al., 2012*), a transcriptional reporter of neuronal activity, to monitor the activity of L-NSC[ITP] in flies exposed to different contexts that challenge their osmotic homeostasis. In agreement with our prediction, L-NSC[ITP] are more active (indicated by increased GFP immunofluorescence) in both males (*Figure 8A*) and females (*Figure 8B*) that were desiccated compared to flies that were kept under normal conditions. Moreover, L-NSC[ITP] activity returned to normal levels in rehydrated flies that were previously exposed to desiccation (*Figure 8A and B*). In order to confirm that increased neuronal activity translates into increased peptide release, we independently quantified ITPa immunofluorescence in different subsets of ITPa-producing brain neurons (*Figure 8C and D*). We observed

reduced fluorescence in ITPa-expressing NSC as well as clock neurons in both males (*Figure 8C*) and females (*Figure 8D*) that had been exposed to desiccation stress. ITPa immunofluorescence returned to normal levels in desiccated flies that were then allowed to rehydrate. Since *ITP* mRNA is upregulated during desiccation, reduced immunofluorescence indicates increased release and not decreased peptide synthesis. Hence, not only do the L-NSC$^{ITP}$ release ITPa into the circulation during desiccation, but the 5$^{th}$-LN$_v$ and LN$_d$$^{ITP}$ likely release ITPa within the brain to modulate other circuits during desiccation. Together, these results demonstrate that ITP neurons are activated and release ITPa during desiccation.

## *ITP* knockdown in ITP-RC-T2A-GAL4 cells impacts osmotic and metabolic homeostasis

Insect ITP is evolutionarily related to CHH, which, as its name indicates, regulates glucose homeostasis in crustaceans (*Chen et al., 2020*). A previous study employing ubiquitous ITP knockdown and overexpression suggests that *Drosophila* ITP also regulates feeding and metabolic homeostasis (*Gáliková and Klepsatel, 2022*) in addition to osmotic homeostasis (*Gáliková et al., 2018*). However, given the nature of the genetic manipulations (ectopic ITPa overexpression and knockdown of *ITP* in all tissues) utilized in those studies, it is difficult to parse the effects of ITP signaling from ITPa-producing neurons. To fill this gap and understand the role of ITP signaling in regulating osmotic

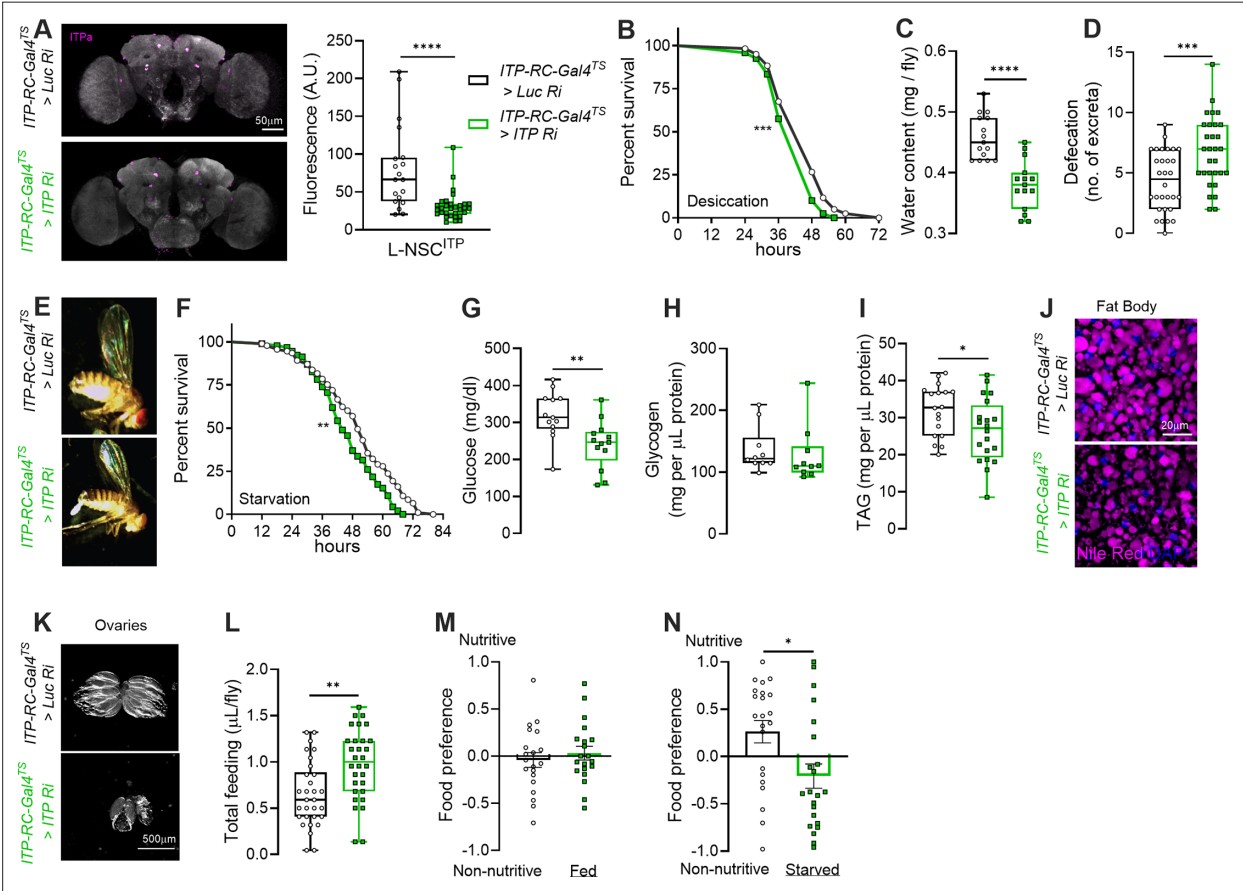

**Figure 9.** Knockdown of ion transport peptide (*ITP*) in adult female *Drosophila* impacts metabolic homeostasis, feeding, and associated behaviors. (**A**) ITPa immunofluorescence is reduced in the L-NSC$^{ITP}$ neurons of flies in which *ITP* was knocked down using *ITP-RC-GAL4$^{TS}$* (*ITP-RC-T2A-GAL4* combined with temperature-sensitive tubulin-GAL80). Flies with *ITP* knockdown are (**B**) less resistant to desiccation tolerance, (**C**) have reduced water content, (**D**) increased defecation, and (**E**) shrunken abdomen. *ITP* knockdown flies (**F**) survive less under starvation, (**G**) have lower levels of circulating glucose, and (**H**) unaffected glycogen levels. However, reduced ITP signaling results in (**I and J**) less lipid levels (TAG = triacylglyceride), and (**K**) smaller ovaries. Moreover, *ITP* knockdown flies exhibit (**L**) increased feeding (over 24 hr) and (**M and N**) defects in preference for nutritive sugars when starved for 18 hr prior to testing. Abbreviations: *Luc Ri, luciferase RNAi; ITP Ri, ITP RNAi*. For (**B and F**), **$p<0.01$, ***$p<0.001$, as assessed by Log-rank (Mantel-Cox) test. For all others, *$p<0.05$, **$p<0.01$, and ****$p<0.0001$ as assessed by unpaired *t*-test.

and metabolic homeostasis in vivo, we specifically knocked down *ITP* using the *ITP-RC-T2A-GAL4*, which includes all the ITPa-expressing neurons. To avoid any developmental effects, we combined temperature-sensitive tubulin GAL80 with the GAL4 line (here referred to as *ITP-RC-GAL4^TS*) to restrict *ITP* knockdown specifically to the adult stage. For these experiments, both the *ITP-RNAi* and the control *luciferase RNAi* lines were first backcrossed for five generations into the wild-type background to minimize genetic background effects. We first successfully confirmed the effectiveness of *ITP-RNAi* by quantifying ITPa immunofluorescence in the brains of control and *ITP* knockdown flies (*Figure 9A*). In agreement with the anti-diuretic effects of ITPa ex vivo, *ITP* knockdown resulted in reduced desiccation tolerance (*Figure 9B*). This is likely a result of reduced water retention (*Figure 9C*) and increased defecation (*Figure 9D*). This reduced water content was also evident from the visibly shrunken abdomens of *ITP* knockdown females compared to controls (*Figure 9E*).

Beyond its effects on osmotic balance, *ITP* knockdown in *ITP-RC* neurons also compromised metabolic homeostasis. Females with *ITP* knockdown exhibited reduced survival under starvation (*Figure 9F*), prompting us to assess their circulating and stored macronutrients. As expected based on lowered starvation tolerance, these flies displayed significantly lower circulating glucose (*Figure 9G*) and lipid stores in the fat body (*Figure 9I and J*). Glycogen levels, however, remained unaltered compared to controls (*Figure 9H*). As a consequence of these depleted energy reserves, the size of the ovaries was reduced in *ITP* knockdown females (*Figure 9K*). Surprisingly, the reduction in energy reserves was not attributable to decreased food intake. On the contrary, *ITP* knockdown flies consumed more food than controls (*Figure 9L*). Independently, we also assayed the food preference of flies when given a choice between a nutritive sugar and a sweeter non-nutritive sugar, since it can report deficits in mechanisms that monitor internal metabolic state. While fed flies of both the control and experimental genotypes showed no preference (*Figure 9M*), starved control flies exhibited a shift towards caloric nutritive sugars (*Figure 9N*), which reflects their drive to restore energy balance. In contrast, starved *ITP* knockdown flies did not display this shift in preference towards nutritive sugars (*Figure 9N*), indicating a disruption in the integration of internal metabolic cues with taste-driven food selection.

Collectively, these results demonstrate that disrupting ITP signaling from *ITP-RC* neurons leads to profound systemic effects on osmotic regulation, metabolic balance, and feeding behaviors, underscoring the pivotal role of ITP in coordinating diverse physiology and behaviors.

## ITPa overexpression in ITP-RC-T2A-GAL4 cells modulates osmotic and metabolic homeostasis

The *ITP-RNAi* used here targets all *ITP* isoforms. Hence, the phenotypes observed following *ITP* knockdown cannot directly be attributed to ITPa. Therefore, we complemented these analyses by specifically overexpressing *ITPa* in adult females using *ITP-RC-GAL4^TS* (*Figure 10*). We first confirmed that driving *UAS-ITPa* with *ITP-RC-GAL4^TS* indeed results in increased ITPa peptide levels in L-NSC^ITP (*Figure 10A*). In agreement with our ex vivo secretion data, ITPa overexpression improves desiccation tolerance (*Figure 10B*), likely due to increased water retention, since flies overexpressing ITPa had higher body water content (*Figure 10C*) and bloated abdomens (*Figure 10D*). Independently, we also assessed recovery from chill-coma as an indirect measure of flies' ionoregulatory capacity (*MacMillan et al., 2012*). ITPa overexpression had no impact on chill coma recovery and tolerance to salt stress (*Figure 10—figure supplement 1*). With regard to metabolic physiology, flies with ITPa overexpression survive longer under starvation (*Figure 10E*). These flies had reduced glucose levels (*Figure 10F*) but their glycogen levels were unaltered (*Figure 10G*). ITPa overexpression also led to increased lipid levels (*Figure 10H*). In addition, flies with ITPa overexpression showed defects in preference between nutritive versus non-nutritive sugar (*Figure 10I and J*) and had larger ovaries (*Figure 10K*). Thus, ITPa overexpression largely results in opposite phenotypes compared to those seen following *ITP* knockdown.

Independently, since ITPa is released from both clock neurons and L-NSC^ITP during desiccation, we asked whether ITPa overexpression in both neuron types affected rhythmic locomotor activity under normal and desiccating conditions. We did not observe any drastic differences in locomotor activity of flies kept under normal conditions and subsequently transferred to empty vials (desiccation conditions) (*Figure 10L*). Therefore, the function of ITPa released from clock neurons during desiccation remains to be determined.

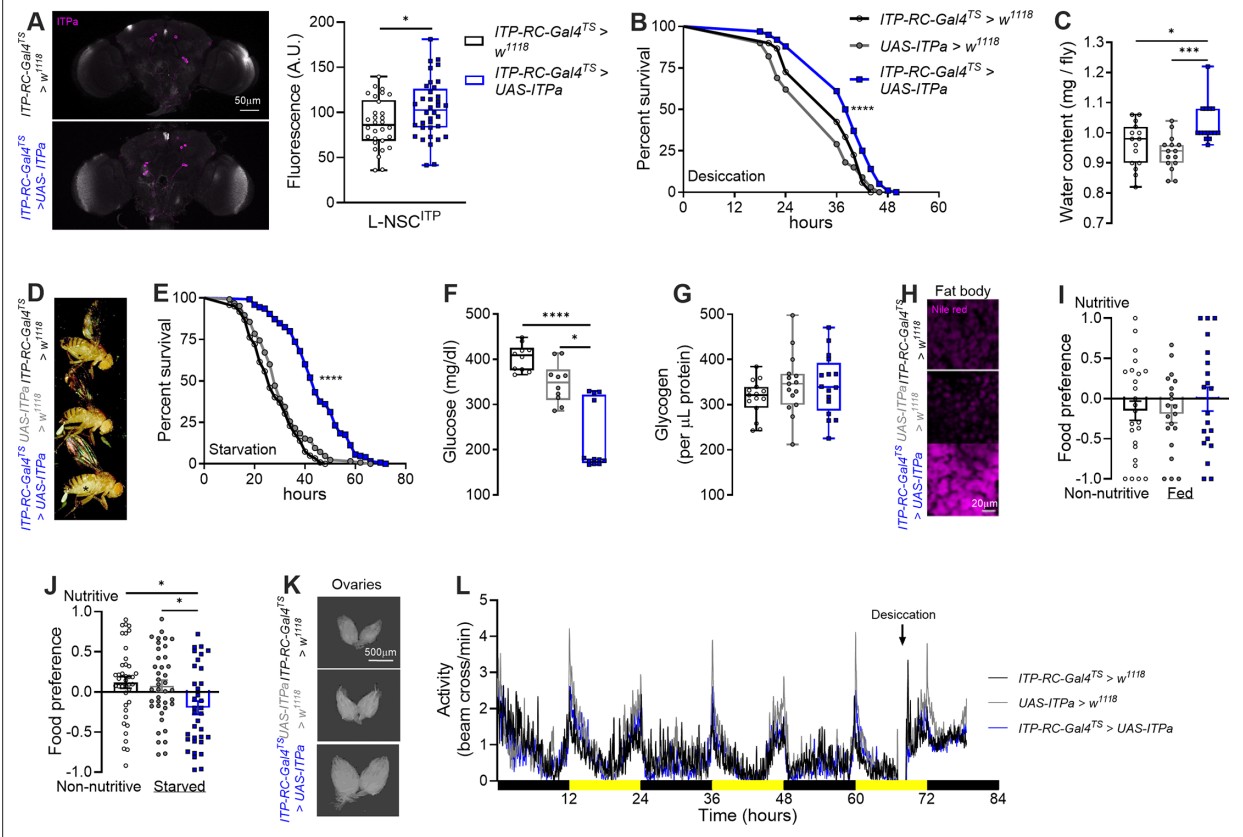

**Figure 10.** ITP amidated (ITPa) overexpression in adult female *Drosophila* impacts osmotic and metabolic homeostasis, feeding, and related behaviors. (**A**) Overexpression of *ITPa* using *ITP-RC-GAL4^TS* results in increased ITPa immunofluorescence in L-NSC^ITP. ITPa overexpression results in (**B**) increased desiccation tolerance, (**C**) increased water content, and (**D**) a slightly bloated abdomen (marked by an asterisk). ITPa overexpression causes (**E**) increased starvation tolerance, (**F**) reduced circulating glucose levels but has no effect on (**G**) glycogen levels. (**H**) The size of neutral lipid droplets (stained with Nile red) is increased in flies with ITPa overexpression. (**I** and **J**) These flies also exhibit defects in preference for nutritive sugars when starved for 16 hr prior to testing. (**K**) ITPa overexpression flies have enlarged ovaries. (**L**) ITPa overexpression has no effect on locomotor activity under fed or desiccating conditions. Black bars indicate night-time and yellow bars indicate daytime. All experiments were performed at 29 °C. For (**B and E**), ****$p<0.0001$, as assessed by Log-rank (Mantel-Cox) test. For (**A**), *$p<0.05$ as assessed by unpaired *t* test. For all other experiments, *$p<0.05$, ***$p<0.001$, ****$p<0.0001$ as assessed by one-way ANOVA followed by Tukey's multiple comparisons test. For clarity, significant pairwise differences compared to only the experimental treatment are indicated.

The online version of this article includes the following figure supplement(s) for figure 10:

**Figure supplement 1.** ITP amidated (ITPa) overexpression in adult female *Drosophila* has no effect on cold and ionic stress.

## ITPa signals via Gyc76C in the renal tubules to modulate osmotic homeostasis

We next explored the role of ITP signaling via Gyc76C in maintaining osmotic homeostasis in vivo. In order to test if the effects on osmotic homeostasis were mediated via Gyc76C in the renal tubules, we monitored osmotic and ionic/salt stress tolerance of flies in which Gyc76C was specifically knocked down in MT principal or stellate cells using the *uro-GAL4* and *c724-GAL4*, respectively (*Figure 11*). As expected, flies with Gyc76C knockdown in the MTs exhibit reduced tolerance to desiccation irrespective of the cell type, principal or stellate, being targeted (*Figure 11A and D*). Interestingly, salt stress impacted flies differently depending on the cell type in which Gyc76C was knocked down. Principal cell knockdown led to increased survival (*Figure 11B*), whereas stellate cell knockdown resulted in reduced survival (*Figure 11E*), which could reflect functional differences between the two cell types. Lastly, Gyc76C knockdown in either MT cell type increased the time taken to recover from chill-coma, highlighting deficits in the ability to maintain ionic homeostasis (*Figure 11C and F*). In summary, ITPa is released into the circulation during desiccation and modulates the MTs to promote tolerance to

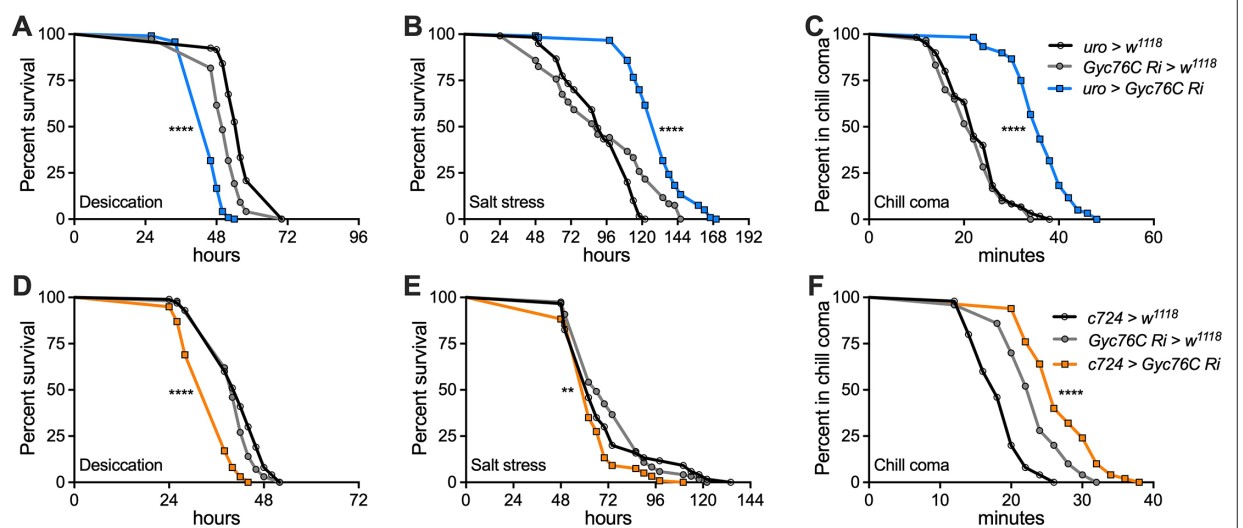

**Figure 11.** Female Malpighian tubule-specific knockdown of *Gyc76C* impacts osmotic and ionic homeostasis. Knockdown of *Gyc76C* in both the (**A**) principal cells of renal tubules using *uro-GAL4* and (**D**) stellate cells using *c724-GAL4* reduces desiccation tolerance. *Gyc76C* knockdown in (**B**) principal cells increases survival under salt stress, whereas knockdown in (**E**) stellate cells lowers survival. (**C** and **F**) *Gyc76C* knockdown in principal or stellate cells increases the time taken for recovery from chill-coma. Abbreviation: *Gyc76C Ri, Gyc76C RNAi*. For all panels, \*\**p*<0.01, \*\*\*\**p*<0.0001, as assessed by Log-rank (Mantel-Cox) test.

osmotic and ionic stresses. This evidence suggests that the hormonal effect of ITPa is likely mediated via Gyc76C expressed in the stellate and principal cells of the MTs.

## ITPa-Gyc76C signaling to the fat body influences metabolic physiology and associated behaviors

Having identified the inter-organ pathway via which ITPa modulates osmotic homeostasis, we next wanted to characterize the pathway regulating metabolic physiology. Since Gyc76C is expressed in the fat body and only the female IPCs, but not in AKH-producing cells, we hypothesized that ITP primarily regulates metabolic homeostasis via direct signaling to the fat body. To test this prediction, we specifically knocked down *Gyc76C* in the female fat body using *yolk-GAL4*. Flies with *Gyc76C* knockdown in the fat body exhibit a drastic reduction in starvation tolerance compared to control flies (*Figure 12A*). Remarkably, these flies start dying after only 4 hr of starvation, whereas control flies can normally tolerate at least 24 hr of starvation. Therefore, we investigated whether Gyc76C signaling to the fat body impacts energy stores. In agreement with reduced starvation survival, *Gyc76C* knockdown flies have lower hemolymph glucose (*Figure 12B*), unaltered glycogen levels (*Figure 12C*), and lower lipids in the fat body (*Figure 12D and E*) compared to controls. Hence, lower lipid levels, especially in the fat body, likely contribute to reduced starvation survival. Interestingly, the reduction in energy stores is not due to decreased food intake because *Gyc76C* knockdown flies fed more than controls (*Figure 12F*). In addition, these flies displayed altered food preferences. Specifically, they preferred yeast over sucrose (*Figure 12G*), possibly to mitigate protein or general caloric deficits since the protein in yeast yields greater caloric value than sugar. Independently, we also assayed the preference of flies for nutritive versus non-nutritive sugars. While there was no preference for nutritive or non-nutritive sugar in fed flies (*Figure 12H*), control flies showed increased preference for nutritive sugar during starvation (*Figure 12I*). Conversely, starved *Gyc76C* knockdown flies displayed a slight preference for non-nutritive sugar over nutritive sugar (*Figure 12I*), suggesting disrupted integration of taste signals with the internal metabolic state. In summary, these experiments indicate that Gyc76C signaling in the fat body is vital in regulating feeding, metabolic homeostasis, and consequently survival.

Next, we examined if fat body-specific *Gyc76C* knockdown impacts other tissues and behaviors. We first observed that *Gyc76C* knockdown flies had drastically shrunken ovaries (*Figure 12J*). In addition, knockdown flies also defecated more than controls (*Figure 12K*) and relatedly had a lower water

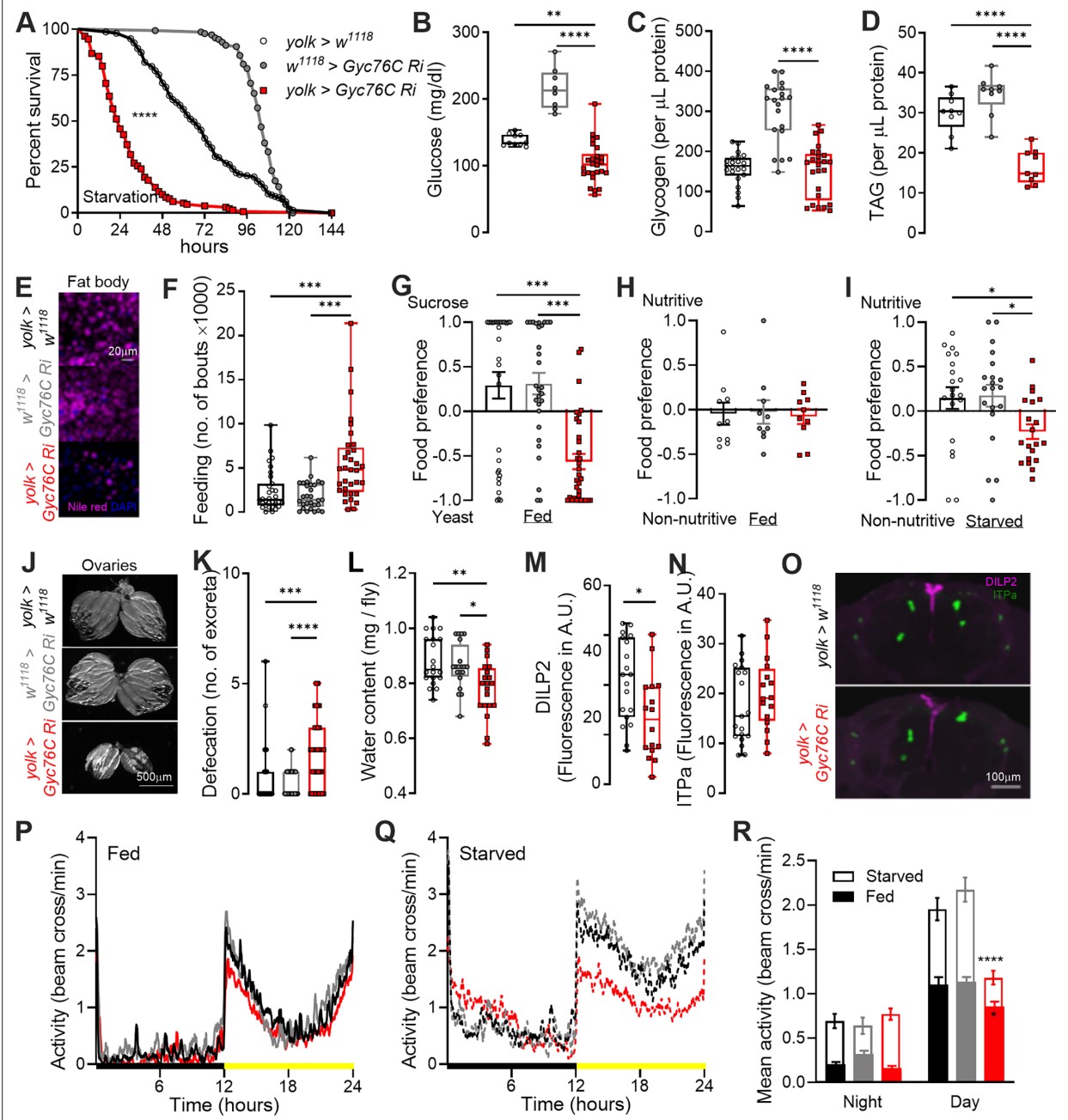

**Figure 12.** *Gyc76C* knockdown in the female fat body using *yolk-GAL4* impacts metabolic homeostasis, feeding, and associated behaviors. Flies with fat body-specific *Gyc76C* knockdown with *UAS-Gyc76C RNAi* (*#106525*) are (**A**) extremely susceptible to starvation and (**B**) have reduced glucose levels. (**C**) Glycogen levels are unaltered in flies with fat body-specific *Gyc76C* knockdown. (**D and E**) However, lipid levels (TAG = triacylglyceride) are drastically reduced. *Gyc76C* knockdown flies exhibit (**F**) increased feeding (over 24 hr), (**G**) a preference for yeast over sucrose, and (**H and I**) defects in preference for nutritive sugars when starved for 4 hr prior to testing. Flies with *Gyc76C* knockdown in the fat body have (**J**) smaller ovaries, they (**K**) defecate more and have (**L**) reduced water content than the controls. For K, the number of excreta counted over 2 hr. *Gyc76C* knockdown also impacts (**M**) DILP2 peptide levels (**N**) but not ITP amidated (ITPa) levels in the neurosecretory cells. CTCF=Corrected Total Cell Fluorescence. (**O**) Representative confocal stacks showing DILP2 and ITPa immunostaining. *Gyc76C* knockdown flies also display reduced daytime locomotor activity under (**P**) fed and (**Q**) and starved conditions compared to controls. Black bars indicate nighttime and yellow bars indicate daytime. (**R**) Average night and daytime activity over one day under fed and starved conditions. For (**A**), ****$p<0.0001$, as assessed by Log-rank (Mantel-Cox) test. For (**M and N**), *$p<0.05$ as assessed by unpaired *t* test. For all others, *$p<0.05$, **$p<0.01$, ***$p<0.001$, ****$p<0.0001$ as assessed by one-way ANOVA followed by Tukey's multiple comparisons test. For clarity, significant pairwise differences compared to only the experimental treatment are indicated.

The online version of this article includes the following figure supplement(s) for figure 12:

*Figure 12 continued on next page*

*Figure 12 continued*

**Figure supplement 1.** *Gyc76C* knockdown with an independent RNAi in females using *yolk-GAL4* impacts stress tolerance, energy stores, and reproductive physiology.

**Figure supplement 2.** *Gyc76C* knockdown with *UAS-Gyc76C Ri* in the female fat body impacts general locomotor activity.

content (*Figure 12L*). These effects could either be caused by altered feeding and metabolism and/ or via an indirect impact on insulin and ITP signaling amongst other pathways. Particularly, reduced insulin and ITP signaling could result in the observed reproductive and excretory phenotypes, respectively. Therefore, we quantified DILP2 and ITPa peptide levels in the brain NSC following knockdown of *Gyc76C* in the fat body. Indeed, knockdown flies have reduced DILP2 peptide levels (*Figure 12M and O*). However, we did not observe any differences in ITPa peptide levels in L-NSC$^{ITP}$ (*Figure 12N and O*). These results suggest that the shrunken ovaries could be directly caused by reduced nutrient stores as well as via an indirect effect on insulin and possibly juvenile hormone signaling since L-NSC$^{DH31}$ innervate the corpora allata. The increased defecation, on the other hand, could likely be a consequence of increased feeding or due to an impact on another osmoregulatory pathway. To further substantiate the fat body-specific role of Gyc76C, we repeated several of the aforementioned analyses using an independent RNAi line (*Gyc76C RNAi #2*) and observed comparable phenotypes. Flies with *Gyc76C* knockdown in the fat body using *Gyc76C RNAi #2* exhibited reduced tolerance to both desiccation and starvation stress (*Figure 12—figure supplement 1*). These flies also displayed reduced lipid stores and smaller ovaries (*Figure 12—figure supplement 1*). Our findings provide strong evidence that disruption of Gyc76C signaling in the fat body exerts profound systemic effects on multiple aspects of physiology and behavior.

Lastly, we monitored general locomotor activity and starvation-induced hyperactivity, the latter of which is largely governed by AKH, insulin and octopamine signaling (*Lee and Park, 2004*; *Yu et al., 2016a*; *Pauls et al., 2021*). Flies with *Gyc76C* knockdown in the fat body displayed reduced daytime activity when kept under either fed or starved conditions for one day (*Figure 12P–R*). Hence, Gyc76C signaling in the fat body does not appear to impact starvation-induced hyperactivity. However, the effect on general locomotor activity led us to examine the activity of fed flies in more detail over a longer time course. For this, we monitored the activity of flies for 10 days under 12:12 hr light/dark cycles and a subsequent 10 days under constant darkness (*Figure 12—figure supplement 2*). While *Gyc76C* knockdown flies displayed reduced activity on day 1, the average activity of these flies over days 2–6 was not significantly different from the controls (*Figure 12—figure supplement 2*). Interestingly, flies with *Gyc76C* knockdown in the fat body appeared to be more sensitive to differences in light cues, showing a strong reduction in locomotor activity only when switched to constant darkness from 12:12 hr light-dark cycles (*Figure 12—figure supplement 2*).

In conclusion, the phenotypes seen following *Gyc76C* knockdown in the fat body largely mirror those seen following *ITP* knockdown in *ITP-RC* neurons, providing further support that ITPa mediates its effects via Gyc76C.

## Synaptic and peptidergic connectivity of ITP neurons

After characterizing the functions of ITP signaling to the renal tubules and the fat body, we wanted to identify pathways regulating ITP signaling and its downstream neuronal targets. To address this, we took advantage of the recently completed FlyWire adult brain connectome (*Dorkenwald al., 2024*; *Schlegel et al., 2024*) to identify pre- and post-synaptic partners of ITP neurons. ITP neurons have a characteristic morphology which was used to identify them in the connectome (*Figure 13A*; *McKim et al., 2024*; *Reinhard et al., 2024*). LN$_d$$^{ITP}$ and 5$^{th}$-LN$_v$ displayed numerous input and output synapses in the brain (*Figure 13B*, *Figure 13—figure supplement 1*). In particular, these neurons have extensive synaptic output in the superior lateral protocerebrum, where Gyc76C-expressing dorsal clock neurons reside (*Figure 5M*, *Figure 5—figure supplement 2*). Consistent with the high number of synapses, both LN$_d$$^{ITP}$ and 5$^{th}$-LN$_v$ receive inputs and provide outputs to a broad range of neurons (*Figure 13C and D*). In addition, both cell types are upstream of at least one pair of DH$_{31}$-expressing NSC (L-NSC$^{DH31}$) (*Figure 13D*). However, it is not yet clear whether these NSCs are the same ones as the L-NSCs$^{DH31}$ that co-express ITPa (*Figure 1B and E*), since there are three pairs of L-NSC$^{DH31}$ in the adult brain (*Reinhard et al., 2024*). Therefore, we did not examine the synaptic connectivity of L-NSC$^{DH31}$ here. Interestingly, LN$_d$$^{ITP}$ receive indirect inputs from VP1l thermo/hygrosensory neurons

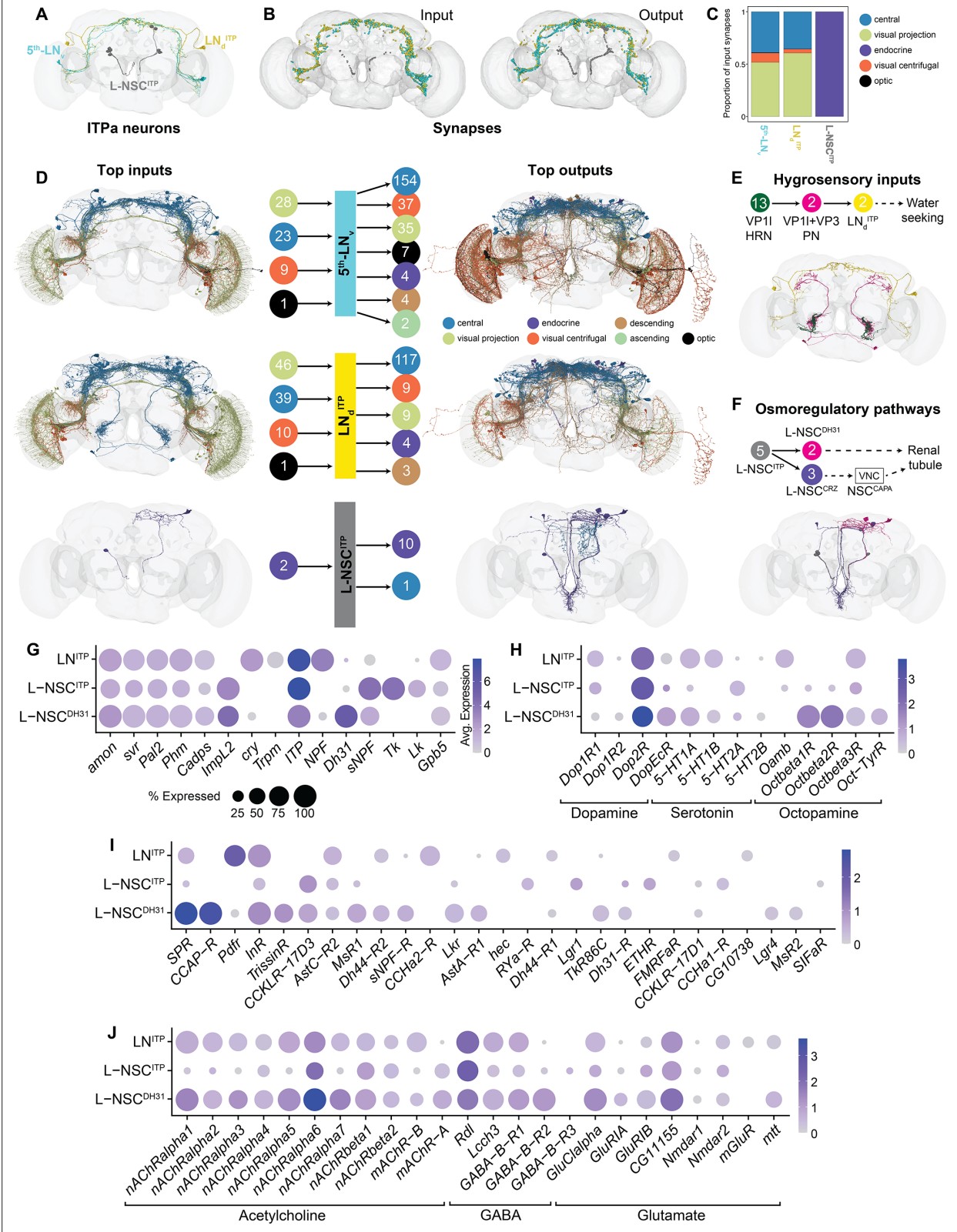

**Figure 13.** Inputs and outputs of ion transport peptide (ITP) neurons based on connectomics and single-cell transcriptomics. (**A**) Reconstruction of ITP amidated (ITPa)-expressing neurons using the complete electron microscopy volume of the adult female brain (data retrieved from the FlyWire platform). Four pairs of lateral neurosecretory cells (L-NSC$^{ITP}$) are gray, fifth ventrolateral neurons (5$^{th}$-LN$_v$) are cyan, and dorsolateral neurons (LN$_d^{ITP}$) are yellow. Diuretic hormone 31 (DH$_{31}$)-expressing lateral neurosecretory cells (L-NSC$^{DH31}$) are not shown since it is unclear which of the three pairs of

Figure 13 continued

L-NSC$^{DH31}$ co-expresses ITPa. (**B**) Location of input and output synapses are colored according to the ITP neuron type. (**C**) Proportion of input synapses (grouped by super class annotations for the FlyWire connectome *Schlegel et al., 2024*) to each ITP neuron type. (**D**) Reconstructions of neurons from different super classes providing inputs to (left) and receiving outputs from (right) 5$^{th}$-LN$_v$, LN$_d^{ITP}$, and L-NSC$^{ITP}$. Only the top 10 cell types are shown here. (Middle) Number of neurons, categorized by super class, providing inputs to and receiving outputs from 5$^{th}$-LN$_v$, LN$_d^{ITP}$, and L-NSC$^{ITP}$. (**E**) Thermo/hygrosensory input pathway to LN$_d^{ITP}$. (**F**) Output from L-NSC$^{ITP}$ to other osmoregulatory hormone-producing cells. (**G**) Identification of single-cell transcriptomes representing different subsets of ITPa-expressing neurons in the adult brain dataset (*Davie et al., 2018*). Since both the 5$^{th}$-LN$_v$ and LN$_d^{ITP}$ co-express *ITP*, *cryptochrome* (*cry*), and *neuropeptide F* (*NPF*), these cells are grouped as LN$^{ITP}$. All three sets of neurons express genes required for neuropeptide processing and release (*amon, svr, Pal2, Phm,* and *Cadps*) and were identified based on the neuropeptides (*ITP, NPF, Dh31, sNPF,* and *Tk*) they express. Dot plots showing expression of (**H**) monoamine, (**I**) neuropeptide, and (**J**) neurotransmitter receptors in different sets of ITPa neurons.

The online version of this article includes the following figure supplement(s) for figure 13:

**Figure supplement 1.** Input and output synapses of ion transport peptide (ITP) neurons.

**Figure supplement 2.** Single-cell transcriptomes of ion transport peptide (*ITP*) neurons in the ventral nerve cord.

(*Figure 13E*; *Choi et al., 2022*). LN$_d^{ITP}$ co-express neuropeptide F (NPF) and could be the same NPF-expressing neurons which have recently been implicated in water seeking during thirst (*Ramirez et al., 2025*). This connectivity could also explain why LN$_d^{ITP}$ show reduced ITPa immunolabeling (indicating ITPa release) following desiccation (*Figure 8C and D*).

In contrast to LN$_d^{ITP}$ and 5$^{th}$-LN$_v$ clock neurons, L-NSC$^{ITP}$ form few significant synaptic connections within this brain volume (*Figure 13B, D, Figure 13—figure supplement 1*). Since these neurons are neurosecretory in nature, their peptides are released from axon terminations in neurohemal areas outside the brain, where regulatory inputs could be located (*Dircksen et al., 2008*; *Kahsai et al., 2010*; *McKim et al., 2024*). It is worth noting that L-NSC$^{ITP}$ output onto other NSC subtypes which secrete osmoregulatory hormones like DH$_{31}$ and corazonin (CRZ) (*Figure 13F*). DH$_{31}$ is a diuretic hormone, whereas CRZ can inhibit CAPA, another diuretic hormone (*Zandawala et al., 2021*).

Since L-NSC$^{ITP}$ receive few synaptic inputs, we hypothesized that their activity, especially during desiccation, is regulated either by cell-autonomous osmosensing or by paracrine and hormonal modulators which transmit the signal from other central or peripheral osmosensors. To address this, we mined single-cell transcriptomes of different subsets of *ITP*-expressing neurons (*Figure 13G*, *Figure 13—figure supplement 2*) from whole brain and VNC datasets (*Davie et al., 2018*; *Allen et al., 2020*) based on markers identified here and previously (*Kahsai et al., 2010*; *Reinhard et al., 2024*). To assess if *ITP*-expressing neurons are cell autonomously osmosensitive, we first examined the expression of transient receptor potential (TRP) and pickpocket (ppk) channels which have been shown to confer osmosensitivity to cells (*Sharif-Naeini et al., 2008*; *Cameron et al., 2010*). Although *Trpm*, a TRP channel, was expressed in LN$^{ITP}$ (*Figure 13G*), we did not detect expression of any TRP or ppk channels in L-NSC$^{ITP}$ and L-NSC$^{DH31}$ (not shown). Thus, unless other (non-characterized) osmosensors are expressed in L-NSC$^{ITP}$, the internal state of thirst/desiccation is likely conveyed to L-NSC$^{ITP}$ via neuromodulators. Intriguingly, the dopamine receptor, *Dop2R*, is highly expressed in all ITP neuron subtypes (*Figure 13H*, *Figure 13—figure supplement 2*). Compared to the *ITP*-expressing neurons in the VNC, the brain neurons seem to be extensively modulated by different neuropeptides (*Figure 13I*, *Figure 13—figure supplement 2*), including those that regulate osmotic homeostasis (diuretic hormone 44, LK, and DH$_{31}$), and feeding and metabolic homeostasis (insulin, Ast-A, CCAP, drosulfakinin, and sNPF). Lastly, with the exception of L-NSC$^{ITP}$, all *ITP* neurons express high levels of neurotransmitter receptors (*Figure 13J*, *Figure 13—figure supplement 2*). This is consistent with fewer synaptic inputs to L-NSC$^{ITP}$. In conclusion, the ITP neuron connectomes and transcriptomes provide the basis to functionally characterize signaling pathways regulating ITP signaling in *Drosophila*.

## Discussion

Insect ITPs are members of the multifunctional family of CHH/MIH neuropeptides that have been intensely investigated in crustaceans for their role in development, reproduction, and metabolism (*Webster et al., 2012*). In *Drosophila,* the three ITP isoforms (ITPa, ITPL1, and ITPL2) were until very recently among the very few neuropeptides whose receptors had not been identified. However, recently an ITPL2-activated GPCR, TkR99D, was identified (*Xu et al., 2023*) similar to the ITPL-activated BNGR-A24 in the moth *Bombyx* (*Nagai et al., 2014*). Thus, receptors for *Drosophila* ITPa

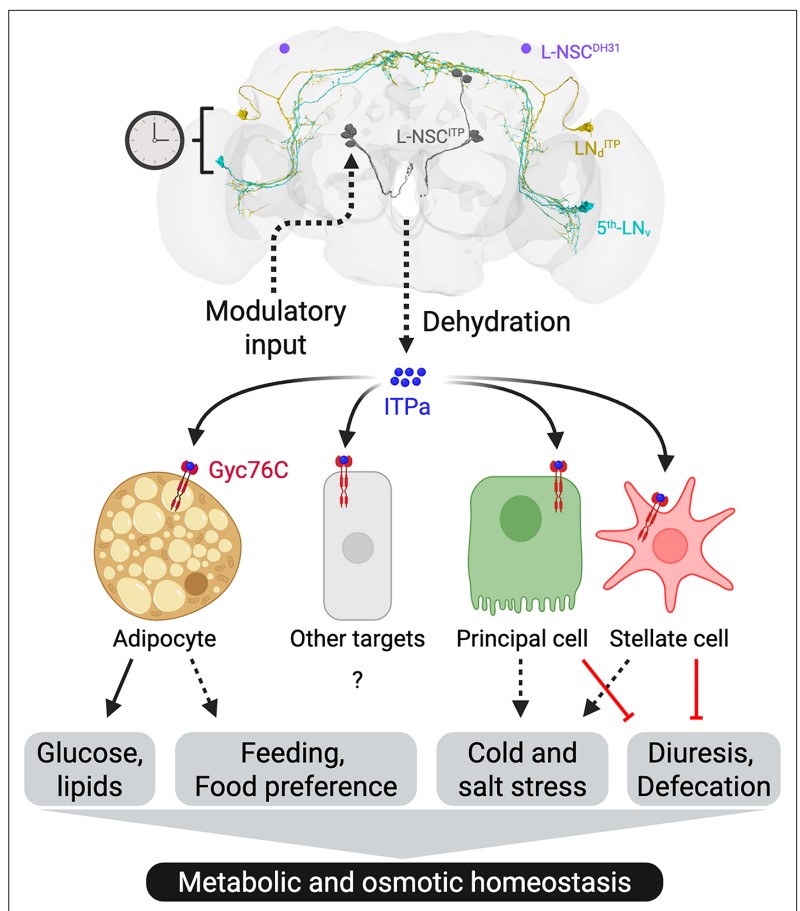

**Figure 14.** A schematic depicting ion transport peptide (ITP) signaling pathways modulating metabolic and osmotic homeostasis in *Drosophila*. Different subsets of ITP neurons in the brain have been color-coded. LN$_d$$^{ITP}$ and 5$^{th}$-LN$_v$ are part of the circadian clock network and regulate clock-associated behaviors and physiology. L-NSC$^{ITP}$ releases ITPa into the circulation following dehydration and information regarding this internal state is likely conveyed to L-NSC$^{ITP}$ by other neuromodulators. Following its release into the hemolymph, ITP amidated (ITPa) activates a membrane guanylate cyclase receptor Gyc76C on the adipocytes in the fat body, principal and stellate cells in the renal tubules, as well as other targets. These signaling pathways affect diverse behaviors and physiology to modulate metabolic and osmotic homeostasis. Dashed arrows depict pathways that remain to be clarified, solid arrows represent direct effects, and red bars represent inhibition. Created with BioRender.com.

and ITPL1 remained to be identified. Furthermore, the neuronal pathways and functional roles of the three ITP isoforms have remained relatively uncharted. Here, using a multipronged approach consisting of anatomical mapping, single-cell transcriptomics, in vitro tests of recombinant ITPa and genetic experiments in vivo, we comprehensively mapped the tissue expression of all three *ITP* isoforms and revealed roles of ITP signaling in regulation of osmotic and metabolic homeostasis via action on MTs and fat body, respectively. We furthermore identified and functionally characterized a receptor for the amidated isoform ITPa, namely the mGC Gyc76C and analyzed its tissue distribution and role in systemic homeostasis. Lastly, we performed connectomics and single-cell transcriptomic analyses to identify synaptic and paracrine pathways upstream and downstream of ITP-expressing neurons. Together, our systematic characterization of ITP signaling establishes a tractable system to decipher how a small set of neurons integrates diverse inputs and orchestrates systemic homeostasis in *Drosophila* (*Figure 14*).

## ITP neurons release multiple neuropeptides to regulate systemic homeostasis

ITPa action on renal tubules and fat body is very likely to be hormonal via the circulation since these tissues are not innervated by neurons. While there are peripheral cells that could possibly release ITPa into the circulation (*Figure 3J*), we consider the eight L-NSC[ITP] to be the major source of hormonal ITPa (and ITPL forms). This is based on the fact that these cells have large cell bodies, numerous dense core vesicles, and extensive axon terminations for production and storage of large amounts of peptide. The smaller L-NSC[DH31] have low levels of ITPa and are likely more suited to locally modulate the corpora allata and/or axon terminations of other *bona fide* NSC in that region. It is noteworthy that L-NSC[ITP] and L-NSC[DH31] express at least five neuropeptides each, with ITPa, ITPL1, sNPF, and glycoprotein hormone beta 5 (Gpb5) being common across both cell types. Additionally, L-NSC[ITP] express ITPL2, LK, and TK, while L-NSC[DH31] express DH$_{31}$. While the functions of *Drosophila* ITPL1 are still unknown, the other neuropeptides have been shown to regulate osmotic and metabolic stress responses (*Johnson et al., 2005*; *Kahsai et al., 2010*; *Zandawala et al., 2018a*; *Díaz-de-la-Peña et al., 2020*). Interestingly, both L-NSC[ITP] and L-NSC[DH31] also express ImpL2, an insulin-binding protein which enables cells to sequester insulin (*Bader et al., 2013*; *Gáliková et al., 2018*; *Ghosh et al., 2022*). These ITP neurons can thus act as a reservoir for insulin-like peptides and could release them along with other neuropeptides to modulate both osmotic and metabolic homeostasis. If we account for DILP2 (*Bader et al., 2013*), L-NSC[ITP] can release up to eight neuropeptides, the most detected for a neuron type so far in *Drosophila*. Hence, understanding the mechanisms by which these cells are regulated can provide novel insights into their orchestrating actions in mediating systemic homeostasis.

Although we consider the L-NSC[ITP] as the main players in hormonal release of ITP isoforms, other ITP-producing neurons could also regulate peripheral tissues. For instance, iag neurons in the abdominal ganglia, which directly innervate the hindgut and rectum, likely modulate gut physiology. These neurons also produce multiple neuropeptides in addition to ITPa, namely ITPL1, CCAP, Ast-A, and Gpb5. CCAP and Ast-A peptides could modulate hindgut contractility (*Vanderveken and O'Donnell, 2014*; *Hillyer, 2018*), whereas ITPa and Gpb5 could regulate water and ion reabsorption (*Sellami et al., 2011*). The concerted action of all these neuropeptides on the hindgut could thus facilitate osmotic homeostasis.

## When is ITPa released and how are ITP neurons regulated?

The release of ITPa from ITPa-expressing neurons in the brain appears to be regulated by the state of water and ion balance in the fly, as seen in our experiments measuring ITP neuron activity and ITPa peptide levels in desiccated and rehydrated flies. But how is this internal state of thirst/desiccation conveyed to ITP neurons? In mammals, osmotic homeostasis is regulated by vasopressin neurons in the hypothalamus (*Voisin and Bourque, 2002*). These neurons monitor changes in the osmotic pressure via intrinsic mechanosensitive channels. In addition, synaptic and paracrine inputs also regulate vasopressin release. Although the vasopressin signaling system has been lost in *Drosophila* (*Nässel and Zandawala, 2019*), other osmoregulatory systems, such as ITP could have evolved similar mechanisms to monitor and consequently regulate the osmotic state of the animal. Our connectomic and single-cell transcriptome analysis indicates that information regarding the osmotic state is likely conveyed to ITP NSC indirectly via one or more neuromodulators released from other osmosensors. Since several receptors for neuromodulators are expressed in L-NSC[ITP], it is difficult to predict which neuromodulators convey the thirst signal to ITP neurons. Nonetheless, it is tempting to speculate that this signal could be dopamine since *Dop2R* is highly expressed in ITP neurons (*Figure 13H*) and dopaminergic neurons also track changes in hydration in mice (*Grove et al., 2022*). Future investigations are needed to explore the modulation of ITP neurons by dopamine and other modulators. Interestingly, the two pairs of clock neurons, 5[th]-LN$_v$ and LN$_d$[ITP], also release ITPa during desiccation (*Figure 8C and D*). The behavioral effects of ITPa signaling by clock neurons during desiccation remains to be discovered, since locomotor activity under normal and desiccation conditions was not affected following ITPa overexpression.

### Additional targets of ITPa-Gyc76C signaling

Gyc76C was identified in *Drosophila* as a mGC in the mid 1990s (*Liu et al., 1995*; *McNeil et al., 1995*), and has since been shown to play diverse roles in embryonic development of different epithelia

(including renal tubules) and muscle (*Patel et al., 2012*; *Patel and Myat, 2013*; *Schleede and Blair, 2015*; *Myat and Patel, 2016*), axonal growth and guidance (*Ayoob et al., 2004*; *Chak and Kolodkin, 2014*), innate immunity (*Iwashita et al., 2020*) and salt stress tolerance (*Overend et al., 2012*). Given the crucial role of Gyc76C during development, it is not surprising that disrupted ITP signaling causes developmental defects in *Tribolium* (*Begum et al., 2009*) and *Drosophila* (McEwan and Zandawala, unpublished). With regards to innate immunity, Gyc76C expression in both the fat body and hemocytes is required for defense against Gram-positive bacteria (*Iwashita et al., 2020*). It remains to be seen if *ITP* knockdown also compromises immunity against Gram-positive bacteria. Besides previously identified functions of Gyc76C, our extensive expression mapping of this receptor also provides insights on other functions of ITPa-Gyc76C signaling. For instance, Gyc76C is expressed in female IPCs (*Figure 5—figure supplement 2*), larval ring gland (*Figure 5—figure supplement 1*) and the adult corpora allata (*Figure 5O*). ITPa-Gyc76C signaling to the IPCs could modulate metabolic physiology associated with female reproduction. Moreover, ITPa-Gyc76C signaling could regulate juvenile hormone signaling in *Drosophila*. It could act similarly to its crustacean homolog, mandibular organ-inhibiting hormone (MOIH), which inhibits secretion of methyl farnesoate, a member of the juvenile hormone family, from the mandibular organs (*Webster et al., 2012*). This could, in turn, impact ovary development and/or vitellogenesis. ITP is also homologous to MIH, which inhibits ecdysteroid production by the Y-organs in crustaceans (*Webster et al., 2012*). Expression of Gyc76C in the larval ring gland, which also includes the ecdysteroid-producing prothoracic glands, suggests that ITP could regulate *Drosophila* development, as shown previously in *Tribolium* (*Begum et al., 2009*). It would also be of interest to determine the functions of Gyc76C in glia, especially those expressing the clock protein, Period (*Figure 5L*). ITPa released by 5th-LN$_v$ and LN$_d$$^{ITP}$ may link the neuronal clock with the clock in glial cells. Future studies could knockdown Gyc76C in these additional targets to identify novel roles of ITPa-Gyc76C signaling in *Drosophila*.

## Functional overlap between mammalian atrial natriuretic peptide (ANP) and *Drosophila* ITP

The multifunctional ITP signaling characterized here is reminiscent of the ANP signaling in mammals (*Komatsu et al., 1991*; *Moro and Smith, 2009*; *Verboven et al., 2017*). ANP is secreted from the cardiac muscle cells to regulate sodium and water excretion by the kidney. Interestingly, ITPL2 is also expressed in the heart muscles (*Figure 3M*) and acts as an anti-diuretic in some contexts (*Xu et al., 2023*). Additional functions of ANP include roles in metabolism, heart function and immune system. Thus, ANP targets white adipocytes to affect lipid metabolism (*Verboven et al., 2017*), similar to the ITPa actions on the *Drosophila* fat body. Furthermore, ANP regulates glucose homeostasis, food intake and pancreatic insulin secretion as shown here for ITPa. A role of ANP as a cytokine in immunity, and with protective effects in tumor growth has also been implicated (*De Vito, 2014*), similar to the cytokine-like action of tumor-derived *Drosophila* ITPL2 (*Xu et al., 2023*). It is interesting to note that ITP and Gyc76C are absent in mammals and no orthologs of ANP have been discovered in invertebrates. While an ortholog of mammalian ANP receptors is present in *Drosophila*, studies characterizing its functions are lacking. It is possible that the ANP system in mammals acquired additional functions that are served by ITP signaling in invertebrates. Functional studies on *Drosophila* ANP-like receptors could shed light on the evolution of these signaling systems.

## Limitations of the study

It is worth pointing out that our phylogenetic analysis identified a second orphan mGC, Gyc32E, as a putative ITPa receptor. Although tissue expression analysis suggests that this receptor is not suited to mediate the osmoregulatory effects of ITPa, we cannot completely rule out the possibility that it also contributes to the metabolic phenotypes of ITPa via actions on IPCs and/or the fat body. It is also of interest to determine whether the three ITP splice forms act in synchrony in cases where they are colocalized in neurons. However, we were unable to determine the specific functions of ITPL1 and ITPL2, as existing RNAi transgenes from *Drosophila* stock centers target all three isoforms. Although ITPL2 functions as an anti-diuretic in a gut tumor model (*Xu et al., 2023*), the functions of ITPL2 released from the nervous system and under normal conditions are still unknown. Recent work in *Aedes aegypti* mosquitoes suggests that ITP and ITPL could have different functions (*Sajadi and Paluzzi, 2024*).

## Concluding remarks

To conclude, our comprehensive characterization of ITP, a homeostatic signaling system with pleiotropic roles, provides a foundation to understand the neuronal and endocrine regulation of thirst-driven behaviors and physiology. L-NSC[ITP], with the potential to release up to eight diverse neuropeptides, likely regulate most aspects of *Drosophila* physiology to modulate systemic homeostasis.

# Materials and methods

## Fly strains

*Drosophila melanogaster* strains used in this study are provided in ***Supplementary file 1***. Unless stated otherwise, flies were raised at 25°C on a standard medium containing 8.0% malt extract, 8.0% corn flour, 2.2% sugar beet molasses, 1.8% yeast, 1.0% soy flour, 0.8% agar, and 0.3% hydroxybenzoic acid. For adult-specific manipulations with *tubulin-GAL80[ts]*, flies were raised at 18°C until two days post-eclosion and then maintained at 29°C until analysis. Unless specified otherwise, all experiments were done using mated females.

## Immunohistochemistry and confocal imaging

Adult *Drosophila* were fixed in 4% paraformaldehyde (PFA) with 0.5% Triton-X100 in 0.1 M sodium phosphate buffer saline (PBST) for 2.5 hr on nutator at room temperature. Larval *Drosophila* were fixed in 4% PFA for 2 hr over ice. After fixation, the flies were washed with 0.5% PBST for 1 hr (4×15 min). Subsequently, the flies were washed with PBS for 10 min. Fixed flies were then dissected in PBS and transferred to tubes containing blocking solution (5% normal goat serum in PBST with sodium azide at 1:100 dilution) on ice. After dissection, tissues were incubated in the primary antibody solution (diluted in blocking solution) for 48 hr at 4°C, followed by four washes with 0.5% PBST (4×15 min), and incubated in secondary antibody (diluted in blocking solution) for 48 hr at 4°C. All the antibodies and fluorophores used in this study are provided in ***Supplementary file 2***. Finally, the flies were washed with 0.5% PBST (3×15 min) followed by washes with PBS (210 min). Samples were mounted using Fluoromount-G (Invitrogen, Thermo Fisher) and imaged with a Leica SPE and TCS SP8 confocal microscopes (Leica Microsystems) using 20 X glycerol, 40 X oil, or 63 X glycerol immersion objectives.

## Fluorescence quantification

Confocal images were processed and the immunofluorescence levels measured using Fiji software. The final immunofluorescence of each sample was calculated by subtracting the background mean intensity from the mean intensity of the desired area.

## Sequence alignments and phylogenetic analysis

BLAST (***Altschul et al., 1990***) and HMMER (***Potter et al., 2018***) searches were performed using the *Drosophila* ITPa prepropeptide sequence to identify ITPL sequences in non-arthropods. ITP prepropeptide sequences were aligned using Clustal Omega (https://www.ebi.ac.uk/Tools/msa/clustalo/) and the conserved residues (at least 70% conservation) shaded using Boxshade (https://junli.netlify.app/apps/boxshade/). Phylogenetic analysis was performed using a custom workflow at NGPhylogeny.fr (***Lemoine et al., 2019***). Briefly, membrane guanylate cyclase receptor protein sequences (accession numbers for the sequences are included in the figure) were aligned using MAFFT (flavor: linsi; gap extension penalty: 0.123; gap opening penalty: 1.53; PAM 250 matrix). The alignment was trimmed using BMGE (BLOSUM 62 matrix; sliding window size: 3; maximum entropy threshold: 0; gap rate cut-off: 0.5; minimum block size: 5). A maximum-likelihood analysis with Smart Model Selection (model selection criteria: AIC; bootstrap: 500; random trees: 5) was used to generate the phylogeny. *Drosophila* guanylyl cyclase alpha and beta subunits were used as outgroups.

## Single-cell transcriptome analysis

Single-nucleus transcriptomes of fat body and Malpighian tubules were mined using the Fly Cell Atlas datasets (***Li et al., 2022***). Single-cell transcriptomes of *ITP*-expressing neurons were mined using the datasets generated earlier (***Davie et al., 2018***; ***Allen et al., 2020***).

The parameters used to identify the different cell types are provided below:

LN (8 cells): ITP > 1 , NPF > 1 , cry > 0 , Phm > 0
L-NSC (7 cells): Tk > 1 , sNPF > 1 , ITP > 1 , ImpL2 > 1 , Crz== 0
L-NSC (6 cells): ITP > 2 , Dh31 > 4 , amon > 0 , Phm > 0
iag (1 cell): AstA > 0 & CCAP > 0 & ITP > 1 & Phm > 0 & amon > 0
non-iag (23 cells): AstA == 0 & CCAP == 0 & ITP > 1 & Phm > 0 & amon > 0

All analyses were performed in R-Studio (v2022.02.0) using the Seurat package (v4.1.1 *Hao et al., 2021*).

## Cell lines

HEK293T cells were used as a heterologous system that does not endogenously express *Drosophila* Gyc76C. Similarly, CHO-K1 cells were used as a heterologous system as it does not endogenously express *Drosophila* PK2-R1 or TkR99D. The murine-derived AtT-20 cell line, which is of neuroendocrine origin from pituitary tumour, was used to heterologously express *Drosophila* ITPa. Cells were purchased from ATCC with STR profiling for cell identity confirmation. Results for microbial contamination, including fungal and mycoplasma species, were negative. Throughout studies, morphological confirmation of cell identity was performed regularly. Mycoplasma status at endpoint was determined to be negative by either chromatin staining or by commercial kit PCR-based screening.

## Recombinant ITPa generation

ITP-PE (ITPa) was amplified from $w^{1118}$ adult mixed-sex whole body cDNA using forward (5'-gcca ccATGTGTTCCCGCAACATAAAGATC-3') and reverse (5'-GCACTTTACTTGCGACCCAGG-3') gene-specific primers and cloned into pGEM T-easy vector and sub-cloned into pcDNA3.1+mammalian expression vector using standard molecular techniques as previously described (*Wahedi and Paluzzi, 2018*). Recombinant ITPa was expressed in AtT-20 cells (ATCC CCL-89), which is a murine-derived cell line of neuroendocrine origin from pituitary tumour, by transfection using Lipofectamine LTX reagent following the manufacturer's protocol. A pcDNA3.1+vector containing mCherry instead of the ITPa construct was used as a control to monitor transfection efficiency. A stable cell line constitutively expressing ITPa was isolated under selection using 600 µg/mL geneticin and scaled up to yield recombinant ITPa for ex vivo Ramsay assay. Heterologous expression of ITPa was verified by immunoblot using a rabbit polyclonal antiserum against the C-terminal region of *Drosophila* ITPa described previously (*Hermann-Luibl et al., 2014*; *Gáliková et al., 2018*) diluted 1:8000 in immunoblot block buffer, whereas E7 beta-tubulin (1:2500) was used as loading control (deposited to the DSHB by Klymkowsky, M.; DSHB Hybridoma Product E7) following a previously described immunoblot protocol (*Rocco and Paluzzi, 2020*). This confirmed ITPa expression in AtT-20 cells, while no such band was detected in mCherry-expressing cells (*Figure 6—figure supplement 2*). Cell lysates were collected and protein samples semi-purified by size-exclusion filtration using centrifugal concentrators with a polyethersulfone membrane (ThermoFisher Scientific, Waltham, MA). Specifically, protein harvested from AtT-20 cells expressing ITPa was centrifuged through 20 kDa molecular weight cut-off (MWCO) concentrators and the flow-through excluding proteins >20 kDa was then transferred to a second centrifugal concentrator with a 5 kDa MWCO. This allowed the expressed ITPa to be concentrated in the retentate since its molecular weight is ~9 kDa and permitted buffer exchange so that the final semi-purified ITPa was reconstituted in 1x phosphate-buffered saline (PBS). The concentration of the semi-purified ITPa was determined by an indirect enzyme-linked immunosorbent assay as previously described (*MacMillan et al., 2018*) using the C-terminal antigen used to generate the ITPa antiserum as a standard.

To improve the purity of heterologously expressed ITPa and to scale up production, recombinant ITPa was independently produced by Genscript (Genscript, Piscataway, NJ) following heterologous expression in proprietary TurboCHO and TurboCHO 2.0 expression systems (Genscript, Piscataway, NJ). To produce C-terminally amidated recombinant ITPa (ITP-PE), human peptidylglycine alpha-amidating monooxygenase was co-expressed along with ITP-PE in the expression system. ITPa included an N-terminal histidine tag that allowed one-step purification following heterologous expression.

## Ex vivo fluid secretion (Ramsay) assay

Fluid secreted by individual MTs was monitored using the classical Ramsay assay (*Ramsay, 1954*), where adult fly MT secretion rates were measured following protocols recently described in detail

(*MacMillan et al., 2018*). Briefly, adult male flies (5-6 days old) were dissected under *Drosophila* saline (*Vanderveken and O'Donnell, 2014*) and the anterior pair of MTs was isolated from the gut at the ureter and then transferred into a 20 µl droplet (comprised of a 1:1 mixture of Schneider's insect medium and *Drosophila* saline) placed over a small well within a Sylgard-lined Petri dish filled with hydrated paraffin oil to prevent sample evaporation. The proximal end of a single MT was pulled out of the bathing droplet and wrapped around a minuten pin so that the ureter was approximately halfway between droplet and the pin. As the MT incubates in the bathing droplet, a secretory droplet forms at the ureter, which, following a 60 min incubation, is then detached and measured using a calibrated eyepiece micrometer. The volume of the secreted fluid is then calculated using the secreted droplet's diameter that allows the fluid secretion rate (FSR) to be determined (FSR = droplet volume/incubation time). To stimulate fluid secretion, diuretic hormones including *Drosophila* leucokinin and $DH_{31}$ were added into the bathing droplet to achieve a final concentration of 10 nM and 1 µM, respectively. Unstimulated tubules were treated with a 1:1 mixture of Schneider's insect medium and *Drosophila* saline alone or with diluted PBS for experiments involving recombinant ITPa.

## GPCR heterologous assay

*Drosophila* GPCRs PK2-R1 (CG8784), and TkR99D (CG7887), which are homologous to *B. mori* ITPa and ITPL receptors, respectively (*Nagai et al., 2014*), were amplified using gene-specific primers as previously described (*Park et al., 2002*; *Birse et al., 2006*) and sub-cloned into the $pcDNA3.1^+$ using standard molecular biology techniques. Receptors were expressed in CHO-K1 cells stably expressing aequorin (CHOK1-aeq), a calcium-activated bioluminescent protein (*Sajadi et al., 2020*). At 48 hr post-transfection with either PK2-R1 or TkR99D, CHOK1-aeq cells were prepared for the heterologous functional assay by resuspension in BSA assay media (DMEM-F12 media containing 0.1% bovine serum albumin (BSA), 1X antimycotic-antibiotic) containing 5 µM coelenterazine *h* (Nanolight Technologies, Pinetop, AZ, USA) and incubated with mixing for 3 hr. After this incubation, cells were diluted 10-fold with BSA assay media reducing the concentration of coelenterazine *h* to 0.5 µM and incubating for an additional hour with constant mixing. Cells were then loaded into individual wells of a white 96-well luminescence plate with an automatic injector unit and luminescence was measured for 20 s using a Synergy 2 Multi-Mode Microplate Reader (BioTek, Winooski, VT, USA). Each well of the 96-well plate was pre-loaded with candidate ligands (recombinant ITPa, pyrokinin 2, and tachykinin 1) at 100 nM and 500 nM (*Park et al., 2002*; *Birse et al., 2006*). BSA assay media alone was utilized as a negative control, while 50 µM ATP, which acts on endogenously expressed purinoceptors (*Iredale and Hill, 1993*), was used as a positive control.

## Gyc76C characterization in HEK293T cells

*Drosophila* Gyc76C, codon-optimized for mammalian expression, was custom-synthesized and sub-cloned into pCDNA3.1(+)-C-HA by Genscript (Piscataway, NJ). HEK293T cells were cultured in Dulbecco's modified Eagle's medium (Corning) supplemented with 10% fetal bovine serum (GenClone, El Cajon, CA) at 37°C and 5% $CO_2$. Cells plated in 35 mm glass bottom imaging dishes (Cellvis, Mountain View, CA) were transfected at 70–80% confluency with 1.5-1.75 µg Gyc76C_pCDNA3.1(+)-C-HA and/or 0.75-1.0 µg Green cGull (*Matsuda et al., 2017*) using Mirus TransIT-LT1 (Mirus Bio, Madison, WI) following the manufacturer's protocol. Cells were then incubated at 30.5°C and 5% $CO_2$ for 24 hr. For imaging, HEK293T media was replaced with modified Ringer's buffer (140 mM NaCl, 3.5 mM KCl, 0.5 mM $NaH_2PO_4$, 0.5 mM S-3 $MgSO_4$, 1.5 m M $CaCl_2$, 10 mM HEPES, 2 mM $NaHCO_3$, and 5 mM glucose) 30 min before imaging. Live-cell images were acquired every 15 s for 6 min using a Stellaris X8 confocal microscope with an 86X water objective. After 90 s of baseline recording, recombinant ITP-PE/PAM (Genscript) was added at 50 nM, 250 nM, or 500 nM final concentration.

Image analysis was performed using FIJI (ImageJ) software. Regions of interest (ROIs) were drawn around cells expressing Green cGull and fluorescence intensity was measured at every timepoint through the stack. The same ROIs were then moved to off-cell regions of the image and background measurements were taken at every timepoint. Background was subtracted from cell measurements in Excel. Baseline was set to the average of the first seven timepoints and all subtracted measurements were divided by this value. Replicates were compiled in GraphPad Prism 10 software for visualization. Area under the curve was computed as the sum of fluorescence intensity over baseline for each cell and then compiled using Prism 10 for visualization. Statistical analysis was performed using

nonparametric distribution one-way ANOVA without matching. Post-hoc multiple comparisons were corrected and subjected to Dunn's test.

To assess Gyc76C expression in imaged cells, imaging dishes were fixed in 4% PFA and subsequently stained with HA-Tag Rabbit mAb (1:2000; Cell Signaling Technology, Danvers, MA) and anti-GFP Goat pAb (1:1000; Rockland). Secondary antibodies (all at 1:1000) included donkey-anti-goat highly cross-absorbed IgG Alexa Fluor 488 (Invitrogen, Carlsbad, CA) or donkey anti-goat IgG Star Green (Abberior, Göttingen, DE) and donkey anti-rabbit highly cross-absorbed IgG Alexa Fluor 647 (Invitrogen, Carlsbad, CA). Fixed cells were imaged on a Keyence BzX-710 microscope with a 20X objective. Images were processed using FIJI (ImageJ) software. Correlations between stained signal intensity and peak live Green cGull fluorescence were performed using Prism 10.

### Feeding assays

flyPAD (*Itskov et al., 2014*) was used to calculate the number of feeding bouts over 24 hr as well as preference between sucrose versus yeast. Individual flies were mouth-pipetted to a flyPAD unit and were given a choice between sucrose (5 mM) and yeast (10%) in 2% agarose. The data was analyzed using a custom script provided by the manufacturer. Total food intake over 24 hr was calculated by adding the number of feeding bouts on sucrose and yeast. Food preference for each fly was calculated by dividing the difference in sucrose and yeast uptake with total food intake.

CAFE assay was performed to monitor the preference between nutritive and non-nutritive sugars. For each genotype, 10 flies (fed or starved) were transferred to an empty glass vial and given a choice between 25 mM D-fructose (nutritive) and 80 mM D-arabinose (non-nutritive) using 5 μL capillaries. Glass vials containing food capillaries without flies were used as controls to monitor evaporation. All the glass vials were maintained in moist chambers and reduction in the volume of individual capillary was measured after 2 hr of feeding. Food preference was calculated as above based on 20 replicates for each genotype.

For experiments involving *ITP* knockdown, total feeding was analysed using the CAFE assay. Ad libitum-fed female flies were anesthetized with $CO_2$ and individually transferred into 2ml tubes. Each fly was provided a 5 μl glass capillary filled with a solution containing 10% sucrose, 10% yeast and 0.1% propionic acid. Flies were allowed to feed for 24 hr before measuring the volume they consumed. To reduce evaporation, all tubes were kept in a humidified box at 25°C on a 12:12 light-dark cycle.

### Glucose assay

Thorax of around 40-50 adult flies were punctured using a 0.1 mm metallic needle. The flies were then transferred to 0.5 ml tubes with a hole at the bottom. These tubes containing the flies were placed inside a 1.5 ml tube and centrifuged for 10 min at 5000 RPM at 4°C. The clear hemolymph collected in the 1.5 ml tubes was used to measure glucose concentration as per the manufacturer's recommended protocol (Glucose calorimetric assay kit, Cayman #10009582). A minimum of 10 replicates were analyzed for each genotype.

### Triglyceride assay

To quantify total triglycerides, five flies for each genotype were homogenized and processed as per the manufacturer's protocol (Triglyceride Calorimetric assay kit, Cayman #10010303). Triglyceride levels were normalized by the protein content. A minimum of 9 replicates were analyzed for each genotype.

### Glycogen assay

To quantify the amount of stored glycogen, five flies for each genotype were homogenized and processed as per the manufacturer's protocol (Glycogen assay kit, Cayman #700480). The amount of glycogen was normalized by the protein content. A minimum of 10 replicates were analyzed for each genotype.

### Protein content

Protein concentrations were measured using the Bradford reagent (Sigma #B6916). Samples were added at a 1:200 ratio in a 96-well plate and incubated at room temperature for 5 min. Finally, the absorbance was measured at 595 nm using a Magellan Sunrise plate reader.

## Water content

Groups of five flies were anesthetized with $CO_2$, placed into tubes and frozen at –80°C for a few hours. Wet weight of the flies was first determined by weighing them. The flies were then dried at 65°C for 48 hr before determining their dry weight. The water content was calculated by subtracting dry weight from wet weight.

## Defecation assay

Flies were anesthetized with $CO_2$ and individually placed into small glass tubes containing blue food 4% sucrose, 2% agarose, and 1% blue food coloring ('Brilliant blue FCF"') for two overnights at 25°C. Afterwards, individual flies were flipped into empty glass tubes, which were placed horizontally in a box at 25°C. The number of feces droplets in each glass tube were manually counted at 2-hr intervals for up to 6 hr.

## *Drosophila* activity monitoring (DAM) experiments

To monitor the locomotor activity of individual flies, *Drosophila* activity monitoring system (Trikinetics Inc, Waltham, Massachusetts) was used. Individual flies were transferred to a thin glass tube (length 5 cm, diameter 5 mm) containing 2% agar and 4% sucrose for fed conditions, 2% agar for starved conditions or left empty for desiccating conditions. Activity was recorded in 1-min intervals under 12:12 light-dark cycles for 8–10 days followed by 8-10 days of constant darkness. The light-dark cycles were maintained using the LED light sources set at 100 lux, housed in a chamber maintained at constant temperature and 70% relative humidity ±5%. The data was analyzed using Actogram J in Fiji (*Schmid et al., 2011*). All analyses were based on approximately 30 flies per genotype.

To assess the impact of desiccation on locomotor activity, locomotor activity of individual flies was recorded in tubes containing 2% agar and 4% sucrose for approximately 66 hr. Following this time, flies were transferred to empty tubes to assess the impact of desiccation on locomotor activity.

## Stress tolerance assays

To monitor starvation survival, flies were individually placed in glass tubes containing 2% agar and their survival estimated automatically (based on lack of activity) using the DAM system as above. To monitor survival under desiccation, groups of 20 flies were kept in empty vials without access to any water or food. Dead flies were quantified visually at regular intervals during daytime. Survival curves were generated based on at least 120 flies per genotypes. Tolerance to salt stress was monitored by maintaining groups of 20 flies each on an artificial diet (medium containing 100 g/L sucrose, 50 g/L yeast, 12 g/L agar, 3 ml/L propionic acid, and 3 g/L nipagin) supplemented with 4% NaCl. Number of dead flies were quantified visually at regular intervals during daytime. Survival curves were generated based on at least 120 flies per genotype. To assess recovery from chill coma, 10 flies for each genotype were transferred into empty vials and kept in ice-cold water (0°C) for 4 hr to induce immediate chill coma. Following this incubation, the vials were transferred to room temperature and the recovery of flies was monitored visually at 2 min intervals. Approximately 100 flies per genotype were analyzed.

## Ovary imaging

Around 50-60 ovaries of each genotype were fixed, mounted, and imaged using a bright-field microscope.

## Synaptic connectivity analyses and data visualization

ITPa-expressing neurons in the FlyWire brain connectome were identified previously (*McKim et al., 2024*; *Reinhard et al., 2024*). FlyWire cell IDs of identified ITPa neurons are provided in *Supplementary file 3*. We used the v783 snapshot of the FlyWire connectome and its annotations for all the analyses (*Dorkenwald et al., 2024*, *Schlegel et al., 2024*). Connectivity was based on updated synapse predictions (*Yu et al., 2025*). We used a threshold of five synapses to identify significant connections. Connectivity analyses were based on custom scripts generated previously (*McKim et al., 2024*; *Reinhard et al., 2024*). All data for figure visualizations were processed and analyzed in R-Studio (2024.04.2+764). FlyWire neuroglancer was used to visualize neuron reconstructions (*Dorkenwald et al., 2022*).

## Statistical analyses

Unless mentioned otherwise, an unpaired t-test was used for comparisons between two genotypes and one-way analysis of variance (ANOVA) followed by Tukey's multiple comparisons test for comparisons between three genotypes. The horizontal line in box-and-whisker plots represents the median. Log-rank (Mantel-Cox) test was used to compare survival and chill coma recovery curves. All statistical analyses were performed using GraphPad Prism and the confidence intervals are included in the figure captions.

## Acknowledgements

The authors would like to thank Irina Wenzel for helpful feedback during the preparation of this manuscript, Dr. Christian Wegener for use of the FlyPad facility, and Dr. Nils Reinhard for assistance with data analyses. We are also thankful to Francesca McEwan for preliminary analyses and to Manpreet Kooner, Selina Hilpert, and Emilia Derksen for technical assistance. We thank the Princeton FlyWire team and members of the Murthy and Seung labs, as well as members of the Allen Institute for Brain Science, for the development and maintenance of FlyWire (supported by BRAIN Initiative grants MH117815 and NS126935 to Murthy and Seung). MZ was supported by funding from the Deutsche Forschungsgemeinschaft (DFG; ZA1296/1-1), and NV INBRE grant from the National Institute of General Medical Sciences (GM103440). JPP was supported by a Natural Sciences and Engineering Research Council of Canada (NSERC) Discovery Grant and an Ontario Ministry of Research Innovation Early Researcher Award. ABB and MHO were supported by a NIGMS COBRE award P20GM130459. SK was supported by JSPS KAKENHI (20H03246). DRN was supported by funding from the Swedish Research Council (Grant Number: 2015–04626). FS was supported by NSERC CGS-D. MA was supported by NSERC PGS-D. JG was supported by funding from the University of Würzburg. We also acknowledge funding from the DFG for the Leica TCS SP8 microscope (251610680, INST 93/809–1 FUGG).

## Additional information

### Competing interests

Meet Zandawala: Reviewing editor, eLife. The other authors declare that no competing interests exist.

### Funding

| Funder | Grant reference number | Author |
| --- | --- | --- |
| Deutsche Forschungsgemeinschaft | ZA1296/1-1 | Meet Zandawala |
| National Institute of General Medical Sciences | GM103440 | Meet Zandawala |
| Natural Sciences and Engineering Research Council of Canada | Discovery Grant | Jean-Paul Paluzzi |
| Ontario Ministry of Research and Innovation | Early Researcher Award | Jean-Paul Paluzzi |
| National Institute of General Medical Sciences | P20GM130459 | Mitchell H Omar |
| Japan Society for the Promotion of Science | 20H03246 | Shu Kondo |
| Vetenskapsrådet | 2015-04626 | Dick R Nässel |
| Natural Sciences and Engineering Research Council of Canada | CGS-D | Farwa Sajadi |
| Natural Sciences and Engineering Research Council of Canada | PGS-D | Marishia Agard |

| Funder | Grant reference number | Author |
|---|---|---|
| University of Würzburg | | Jayati Gera |

The funders had no role in study design, data collection and interpretation, or the decision to submit the work for publication.

## Author contributions

Jayati Gera, Data curation, Formal analysis, Supervision, Investigation, Visualization, Methodology, Writing – review and editing; Marishia Agard, Austin B Baldridge, Farwa Sajadi, Leena Thorat, Formal analysis, Investigation, Methodology, Writing – review and editing; Hannah Nave, Formal analysis, Investigation, Methodology, Writing – review and editing, Supervision; Theresa H McKim, Data curation, Investigation, Visualization, Methodology, Writing – review and editing, Computational analysis; Shu Kondo, Resources, Funding acquisition, Writing – review and editing; Dick R Nässel, Conceptualization, Supervision, Funding acquisition, Writing – original draft, Project administration, Writing – review and editing; Mitchell H Omar, Formal analysis, Supervision, Funding acquisition, Investigation, Visualization, Methodology, Writing – review and editing; Jean-Paul Paluzzi, Conceptualization, Formal analysis, Supervision, Funding acquisition, Investigation, Visualization, Methodology, Writing – original draft, Project administration, Writing – review and editing; Meet Zandawala, Conceptualization, Data curation, Formal analysis, Supervision, Funding acquisition, Investigation, Visualization, Methodology, Writing – original draft, Project administration, Writing – review and editing

## Author ORCIDs

Jayati Gera https://orcid.org/0000-0002-4744-1546
Marishia Agard https://orcid.org/0009-0009-1463-6409
Theresa H McKim https://orcid.org/0000-0002-8501-6487
Shu Kondo https://orcid.org/0000-0002-4625-8379
Dick R Nässel https://orcid.org/0000-0002-1147-7766
Mitchell H Omar https://orcid.org/0000-0002-1713-4473
Jean-Paul Paluzzi https://orcid.org/0000-0002-7761-0590
Meet Zandawala https://orcid.org/0000-0001-6498-2208

Reviewer #1 (Public review): https://doi.org/10.7554/eLife.97043.3.sa1
Reviewer #2 (Public review): https://doi.org/10.7554/eLife.97043.3.sa2
Reviewer #3 (Public review): https://doi.org/10.7554/eLife.97043.3.sa3
Author response https://doi.org/10.7554/eLife.97043.3.sa4

# Additional files

## Supplementary files

MDAR checklist

Supplementary file 1. Fly strains used in this study.

Supplementary file 2. Antibodies used for immunohistochemistry.

Supplementary file 3. Root IDs (v783) of ITPa-producing cells in the FlyWire connectome.

## Data availability

Custom code used for connectome and single-cell transcriptome analyses is available at: https://github.com/Zandawala-lab/Gera-et-al-2025-Drosophila-ITPa-Gyc76C, copy archived at *Zandawala and McKim, 2025*. Custom code and output files are available at: https://doi.org/10.5281/zenodo.16747141.

The following dataset was generated:

| Author(s) | Year | Dataset title | Dataset URL | Database and Identifier |
|---|---|---|---|---|
| McKim T, Zandawala M | 2025 | Gera et al 2025: *Drosophila* ITPa-Gyc76C | https://doi.org/10.5281/zenodo.16747141 | Zenodo, 10.5281/zenodo.16747141 |

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
