## [Editor Report · eLife Assessment]

The authors used comprehensive approaches to identify Gyc76C as an ITPa receptor in *Drosophila*. They revealed that ITPa acts via Gyc76C in the renal tubules and fat body to modulate osmotic and metabolic homeostasis. The designed experiments, data, and analyses **convincingly** support the main claims. The findings are **important** to help us better understand how ITP signals contribute to systemic homeostasis regulation.

---

## [Referee Report · Reviewer #1 (Public review)]

Summary:

In *Drosophila melanogaster*, ITP has functions in feeding, drinking, metabolism, excretion, and circadian rhythm. In the current study, the authors characterized and compared the expression of all three ITP isoforms (ITPa and ITPL1&2) in the CNS and peripheral tissues of Drosophila. An important finding is that they functionally characterized and identified Gyc76C as an ITPa receptor in Drosophila using both in vitro and in vivo approaches. In vitro, the authors nicely confirmed that the inhibitory function of recombinant Drosophila ITPa on MT secretion is Gyc76C-dependent (knockdown of Gyc76C specifically in two types of cells abolished the anti-diuretic action of Drosophila ITPa on renal tubules). They also confirmed that ITPa activates Gyc76C in a heterologous system. The authors used a combination of multiple approaches to investigate the roles of ITPa and Gyc76C on osmotic and metabolic homeostasis modulation in vivo. They revealed that ITPa signaling to renal tubules and fat body modulates osmotic and metabolic homeostasis via Gyc76C.

Furthermore, they tried to identify the upstream and downstream of ITP neurons in the nervous system by using connectomics and single-cell transcriptomic analysis. I found this interesting manuscript to be well-written and described. The findings in this study are valuable to help understand how ITP signals work on systemic homeostasis regulation. Both anatomical and single-cell transcriptome analysis here should be useful to many in the field.

Strengths:

The question (what receptors of ITPa in Drosophila) that this study tries to address is important. The authors ruled out the Bombyx ITPa receptor orthologs as potential candidates. They identified a novel ITP receptor by using phylogenetic, anatomical analysis, and both in vitro and in vivo approaches.

The authors exhibited detailed anatomical data of both ITP isoforms and Gyc76C (in the main and supplementary figures), which helped audiences understand the expression of the neurons studied in the manuscript.

They also performed connectomes and single-cell transcriptomics analyses to study the synaptic and peptidergic connectivity of ITP-expressing neurons. This provided more information for better understanding and further study of systemic homeostasis modulation.

Comments on revisions:

In the revised manuscript, the authors addressed all my concerns.

There is one more suggestion: The scale bar for fly and ovary images should be included in Figures 9, 10, and 12.

---

## [Referee Report · Reviewer #2 (Public review)]

The physiology and behaviour of animals are regulated by a huge variety of neuropeptide signalling systems. In this paper, the authors focus on the neuropeptide ion transport peptide (ITP), which was first identified and named on account of its effects on the locust hindgut (Audsley et al. 1992). Using Drosophila as an experimental model, the authors have mapped the expression of three different isoforms of ITP, all of which are encoded by the same gene.

The authors then investigated candidate receptors for isoforms of ITP. Firstly, Drosophila orthologs of G-protein coupled receptors (GPCRs) that have been reported to act as receptors for ITPa or ITPL in the insect Bombyx mori were investigated. Importantly, the authors report that ITPa does not act as a ligand for the GPCRs TkR99D and PK2-R1. Therefore, the authors investigated other putative receptors for ITPs. Informed by a previously reported finding that ITP-type peptides cause an increase in cGMP levels in cells/tissues (Dircksen, 2009, Nagai et al., 2014), the authors investigated guanylyl cyclases as candidate receptors for ITPs. In particular, the authors suggest that Gyc76C may act as an ITP receptor in Drosophila. Evidence that Gyc76C may be involved in mediating effects of ITP in Bombyx was first reported by Nagai et al. (2014) and here the authors present further evidence, based on a proposed concordance in the phylogenetic distribution ITP-type neuropeptides and Gyc76C and experimental demonstration that ITPa causes dose-dependent stimulation of cGMP production in HEK cells expressing Gyc76C. Having performed detailed mapping of the expression of Gyc76C in Drosophila, the authors then investigated if Gyc76C knockdown affects the bioactivity of ITPa in Drosophila. The inhibitory effect of ITPa on leucokinin- and diuretic hormone-31-stimulated fluid secretion from Malpighian tubules was found to be abolished when expression of Gyc76C was knocked down in stellate cells and principal cells, respectively.

Having investigated the proposed mechanism of ITPa signalling in Drosophila, the authors then investigate its physiological roles at a systemic level. The authors present evidence that ITPa is released during desiccation and accordingly overexpression of ITPa increases survival when animals are subjected to desiccation. Furthermore, knockdown of Gyc76C in stellate or principal cells of Malphigian tubules decreases survival when animals are subject to desiccation. Furthermore, the relevance of the phenotypes observed to potential in vivo actions of ITPa is also explored and publicly available connectomic data and single-cell transcriptomic data are analysed to identify putative inputs and outputs of ITPa expressing neurons.

Strengths of this paper.

(1) The main strengths of this paper are:

(i) the detailed analysis of the expression and actions of ITP and the phenotypic consequences of over-expression of ITPa in Drosophila.

(ii). the detailed analysis of the expression of Gyc76C and the phenotypic consequences of knockdown of Gyc76C expression in Drosophila.

(iii). the experimental demonstration that ITPa causes dose-dependent stimulation of cGMP production in HEK cells expressing Gyc76C, providing biochemical evidence that the effects of ITPa in Drosophila are, at least in part, mediated by Gyc76C.

(2) Furthermore, the paper is generally well written and the figures are of good quality.

Weaknesses of this paper.

A weakness of this paper is the phylogenetic analysis to investigate if there is correspondence in the phylogenetic distribution of ITP-type and Gyc76C-type genes/proteins. Unfortunately, the evidence presented is rather limited in scope. Essentially, the authors report that they only found ITP-type and Gyc76C-type genes/proteins in protostomes, but not in deuterostomes. What is needed is a more fine-grained analysis at the species level within the protostomes. However, I recognise that such a detailed analysis may extend beyond the scope of this paper, which is already rich in data.

---

## [Referee Report · Reviewer #3 (Public review)]

Summary:

The goal of this paper is to characterize an anti-diuretic signaling system in insects using *Drosophila melanogaster* as a model. Specifically, the authors wished to characterize a role for ion transport peptide (ITP) and its isoforms in regulating diverse aspects of physiology and metabolism. The authors combined genetic and comparative genomic approaches with classical physiological techniques and biochemical assays to provide a comprehensive analysis of ITP and its role in regulating fluid balance and metabolic homeostasis in Drosophila. The authors further characterized a previously unrecognized role for Gyc76C as a receptor for ITPa, an amidated isoform of ITP, and in mediating the effects of ITPa on fluid balance and metabolism. The evidence presented in favor of this model is very strong as it combines multiple approaches and employs ideal controls. Taken together, these findings represent an important contribution to the field of insect neuropeptides and neurohormones and has strong relevance for other animals. The authors have addressed all weaknesses raised in my previous review.

---

## [Author Response]

The following is the authors’ response to the current reviews.

**Reviewer #1 (Public review):**
The scale bar for fly and ovary images should be included in Figures 9, 10, and 12.

We agree with this comment and apologize for the oversight. We have now modified Figures 9, 10, and 12 to include the scale bars for the ovary images. The fly images were acquired using a stereo microscope where scale bar calculation was not possible. However, all images were acquired at the same magnification for consistency.

**Reviewer #2 (Public review):**
A weakness of this paper is the phylogenetic analysis to investigate if there is correspondence in the phylogenetic distribution of ITP-type and Gyc76C-type genes/proteins. Unfortunately, the evidence presented is rather limited in scope. Essentially, the authors report that they only found ITP-type and Gyc76C-type genes/proteins in protostomes, but not in deuterostomes. What is needed is a more fine-grained analysis at the species level within the protostomes. However, I recognise that such a detailed analysis may extend beyond the scope of this paper, which is already rich in data.

We thank the reviewer for their comment and the suggestion to perform a fine-grained species level comparison of ITP and Gyc76C genes across protostomes. We are unsure of the utility of this analysis for the present study given that we have now shown that ITPa can activate Gyc76C using both an ex vivo and a heterologous assay, the latter being the gold standard in GPCR and guanylate cyclase discovery (see Huang et al 2025 https://doi.org/10.1073/pnas.2420966122; Beets et al 2023 https://doi.org/10.1016/j.celrep.2023.113058); Chang et al 2009 https://doi.org/10.1073/pnas.0812593106.

Additionally, absence of a gene in a genome/proteome is hard to prove especially when many/most of the protostomian datasets are not as high-quality as those of model systems (e.g. *Drosophila melanogaster* and *Caenorhabditis elegans*). Secondly, based on previous findings in *Bombyx mori* (Nagai et al. 2014 https://doi.org/10.1074/jbc.m114.590646 and Nagai et al. 2016 https://doi.org/10.1371/journal.pone.0156501) and *Drosophila* (Xu et al. 2023 https://doi.org/10.1038/s41586-023-06833-8 and our study) it is evident that different products of the ITP gene (ITPa and ITPL) could signal via different receptor types depending on the species. Hence, we would need to explore the presence of several genes (ITP, tachykinin, pyrokinin, tachykinin receptor, pyrokinin receptor, CG30340 orphan receptor and Gyc76C) to fully understand which components of these diverse signaling systems are present in a given species to decipher the potential for cross-talk.

While this species-level comparison will certainly be useful in the context of ITP-Gyc76C evolution, it will not alter the conclusions of the present study – ITPa acts via Gyc76C in *Drosophila*. We therefore agree with the reviewer that these analyses are beyond the scope of this paper.

The following is the authors’ response to the original reviews.

**Reviewer #1 (Public Review):**
Summary:In *Drosophila melanogaster*, ITP has functions on feeding, drinking, metabolism, excretion, and circadian rhythm. In the current study, the authors characterized and compared the expression of all three ITP isoforms (ITPa and ITPL1&2) in the CNS and peripheral tissues of Drosophila. An important finding is that they functionally characterized and identified Gyc76C as an ITPa receptor in Drosophila using both in vitro and in vivo approaches. In vitro, the authors nicely confirmed that the inhibitory function of recombinant Drosophila ITPa on MT secretion is Gyc76C-dependent (knockdown Gyc76C specifically in two types of cells abolished the anti-diuretic action of Drosophila ITPa on renal tubules). They also used a combination of multiple approaches to investigate the roles of ITPa and Gyc76C on osmotic and metabolic homeostasis modulation in vivo. They revealed that ITPa signaling to renal tubules and fat body modulates osmotic and metabolic homeostasis via Gyc76C.Furthermore, they tried to identify the upstream and downstream of ITP neurons in the nervous system by using connectomics and single-cell transcriptomic analysis. I found this interesting manuscript to be well-written and described. The findings in this study are valuable to help understand how ITP signals work on systemic homeostasis regulation. Both anatomical and single-cell transcriptome analysis here should be useful to many in the field.

We thank this reviewer for the positive and thorough assessment of our manuscript.

Strengths:The question (what receptors of ITPa in Drosophila) that this study tries to address is important. The authors ruled out the Bombyx ITPa receptor orthologs as potential candidates. They identified a novel ITP receptor by using phylogenetic, anatomical analysis, and both in vitro and in vivo approaches.The authors exhibited detailed anatomical data of both ITP isoforms and Gyc76C (in the main and supplementary figures), which helped audiences understand the expression of the neurons studied in the manuscript.They also performed connectomes and single-cell transcriptomics analysis to study the synaptic and peptidergic connectivity of ITP-expressing neurons. This provided more information for better understanding and further study on systemic homeostasis modulation.Weaknesses:In the discussion section, the authors raised the limitations of the current study, which I mostly agree with, such as the lack of verification of direct binding between ITPa and Gyc76C, even though they provided different data to support that ITPa-Gyc76C signaling pathway regulates systemic homeostasis in adult flies.

We now provide evidence of Gyc76C activation by ITPa in a heterologous system (new Figure 7 and Figure 7 Supplement 1).

**Reviewer #2 (Public Review):**
Summary:The physiology and behaviour of animals are regulated by a huge variety of neuropeptide signalling systems. In this paper, the authors focus on the neuropeptide ion transport peptide (ITP), which was first identified and named on account of its effects on the locust hindgut (Audsley et al. 1992). Using Drosophila as an experimental model, the authors have mapped the expression of three different isoforms of ITP (Figures 1, S1, and S2), all of which are encoded by the same gene.The authors then investigated candidate receptors for isoforms of ITP. Firstly, Drosophila orthologs of G-protein coupled receptors (GPCRs) that have been reported to act as receptors for ITPa or ITPL in the insect Bombyx mori were investigated. Importantly, the authors report that ITPa does not act as a ligand for the GPCRs TkR99D and PK2-R1 (Figure S3). Therefore, the authors investigated other putative receptors for ITPs. Informed by a previously reported finding that ITP-type peptides cause an increase in cGMP levels in cells/tissues (Dircksen, 2009, Nagai et al., 2014), the authors investigated guanylyl cyclases as candidate receptors for ITPs. In particular, the authors suggest that Gyc76C may act as an ITP receptor in Drosophila.Evidence that Gyc76C may be involved in mediating effects of ITP in Bombyx was first reported by Nagai et al. (2014) and here the authors present further evidence, based on a proposed concordance in the phylogenetic distribution ITP-type neuropeptides and Gyc76C (Figure 2). Having performed detailed mapping of the expression of Gyc76C in Drosophila (Figures 3, S4, S5, S6), the authors then investigated if Gyc76C knockdown affects the bioactivity of ITPa in Drosophila. The inhibitory effect of ITPa on leucokinin- and diuretic hormone-31-stimulated fluid secretion from Malpighian tubules was found to be abolished when expression of Gyc76C was knocked down in stellate cells and principal cells, respectively (Figure 4). However, as discussed below, this does not provide proof that Gyc76C directly mediates the effect of ITPa by acting as its receptor. The effect of Gyc76C knockdown on the action of ITPa could be an indirect consequence of an alteration in cGMP signalling.Having investigated the proposed mechanism of ITPa in Drosophila, the authors then investigated its physiological roles at a systemic level. In Figure 5 the authors present evidence that ITPa is released during desiccation and accordingly, overexpression of ITPa increases survival when animals are subjected to desiccation. Furthermore, knockdown of Gyc76C in stellate or principal cells of Malphigian tubules decreases survival when animals are subject to desiccation. However, whilst this is correlative, it does not prove that Gyc76C mediates the effects of ITPa. The authors investigated the effects of knockdown of Gyc76C in stellate or principal cells of Malphigian tubules on (i). survival when animals are subject to salt stress and (ii). time taken to recover from of chill coma. It is not clear, however, why animals overexpressing ITPa were also not tested for its effect on (i). survival when animals are subject to salt stress and (ii). time taken to recover from of chill coma. In Figures 6 and S8, the authors show the effects of Gyc76C knockdown in the female fat body on metabolism, feeding-associated behaviours and locomotor activity, which are interesting. Furthermore, the relevance of the phenotypes observed to potential in vivo actions of ITPa is explored in Figure 7. The authors conclude that "increased ITPa signaling results in phenotypes that largely mirror those seen following Gyc76C knockdown in the fat body, providing further support that ITPa mediates its effects via Gyc76C." Use of the term "largely mirror" seems inappropriate here because there are opposing effects- e.g. decreased starvation resistance in Figure 6A versus increased starvation resistance in Figure 7A. Furthermore, as discussed above, the results of these experiments do not prove that the effects of ITPa are mediated by Gyc76C because the effects reported here could be correlative, rather than causative.

We thank this reviewer for an extremely thorough and fair assessment of our manuscript.

We have now performed salt stress tolerance and chill coma recovery assays using flies over-expressing ITPa (new Figure 10 Supplement 1).

We agree that the use of the term “largely mirrors” to describe the effects of ITPa overexpression and Gyc76C knockdown is not appropriate and have changed this sentence. We also agree that the experiments did not provide direct evidence that the effects of ITPa are mediated by Gyc76C. To address this, we now provide evidence of Gyc76C activation by ITPa in a heterologous system (new Figure 7 and Figure 7 Supplement 1).

Lastly, in Figures 8, S9, and S10 the authors analyse publicly available connectomic data and single-cell transcriptomic data to identify putative inputs and outputs of ITPa-expressing neurons. These data are a valuable addition to our knowledge ITPa expressing neurons; but they do not address the core hypothesis of this paper - namely that Gyc76C acts as an ITPa receptor.

The goal of our study was to comprehensively characterize an anti-diuretic system in Drosophila. Hence, in addition to identifying the receptor via which ITPa exerts its effects, we also wanted to understand how ITPa-producing neurons are regulated. Connectomic and single-cell transcriptomic analyses are highly appropriate for this purpose. We have now updated the connectomic analyses using an improved connectome dataset that was released during the revision of this manuscript. Our new analysis shows that lNSC^ITP^ are connected to other endocrine cells that produce other homeostatic hormones (new Figure 13F). We also identify a pathway through which other ITP-producing neurons (LNd^ITP^) receive hygrosensory inputs to regulate water seeking behavior (new Figure 13E). Moreover, we now include results which showcase that ITPa-producing neurons (l-NSC^ITP^) are active (new Figure 8A and B) and release ITPa under desiccation. Together with other analyses, these data provide a comprehensive outlook on the when, what and how ITPa regulates systemic homeostasis.

Strengths:(1) The main strengths of this paper are (i) the detailed analysis of the expression and actions of ITP and the phenotypic consequences of overexpression of ITPa in Drosophila. (ii). the detailed analysis of the expression of Gyc76C and the phenotypic consequences of knockdown of Gyc76C expression in Drosophila.(2) Furthermore, the paper is generally well-written and the figures are of good quality.

We thank this reviewer for highlighting the strengths of this manuscript.

Weaknesses:(1) The main weakness of this paper is that the data obtained do not prove that Gyc76C acts as a receptor for ITPa. Therefore, the following statement in the abstract is premature: "Using a phylogenetic-driven approach and the ex vivo secretion assay, we identified and functionally characterized Gyc76C, a membrane guanylate cyclase, as an elusive Drosophila ITPa receptor." Further experimental studies are needed to determine if Gyc76C acts as a receptor for ITPa. In the section of the paper headed "Limitations of the study", the authors recognise this weakness. They state "While our phylogenetic analysis, anatomical mapping, and ex vivo and in vivo functional studies all indicate that Gyc76C functions as an ITPa receptor in Drosophila, we were unable to verify that ITPa directly binds to Gyc76C. This was largely due to the lack of a robust and sensitive reporter system to monitor mGC activation." It is not clear what the authors mean by "the lack of a robust and sensitive reporter system to monitor mGC activation". The discovery of mGCs as receptors for ANP in mammals was dependent on the use of assays that measure GC activity in cells (e.g. by measuring cGMP levels in cells). Furthermore, more recently cGMP reporters have been developed. The use of such assays is needed here to investigate directly whether Gyc76C acts as a receptor for ITPa. In summary, insufficient evidence has been obtained to conclude that Gyc76C acts as a receptor for ITPa. Therefore, I think there are two ways forward, either:(a) The authors obtain additional biochemical evidence that ITPa is a ligand for Gyc76C.or(b) The authors substantially revise the conclusions of the paper (in the title, abstract, and throughout the paper) to state that Gyc76C MAY act as a receptor for ITPa, but that additional experiments are needed to prove this.

We thank the reviewer for this comment and agree with the two options they propose. We had previously tried different a cGMP reporter (Promega GloSensor cGMP assay) to monitor activation of Gyc76C by ITPa in a heterologous system. Unfortunately, we were not successful in monitoring Gyc76C activation by ITPa. We now utilized another cGMP sensor, Green cGull, to show that ITPa can indeed activate Gyc76C heterologously expressed in HEK cells (new Figure 7 and Figure 7 Supplement 1). However, we still cannot rule out the possibility that ITPa can act on additional receptors in vivo. This is based on our ex vivo Malpighian tubule assays (new Figure 6E and F). ITPa inhibits DH31- and LK-stimulated secretion and we show that this effect is abolished in Gyc76C knockdown specifically in principal and stellate cells, respectively. Interestingly, application of ITPa alone can stimulate secretion when Gyc76C is knocked down in principal cells (new Figure 6E). This could be explained by: (1) presence of another receptor for ITPa which results in diuretic actions and/or (2) low Gyc76C signaling activity (RNAi based knockdown lowers signaling but does not abolish it completely) could alter other intracellular messenger pathways that promote secretion. We have added text to indicate the possibility of other ITPa receptors. Nonetheless, our conclusions are supported by the heterologous assay results which indicate that ITPa can activate Gyc76C. Therefore, we do not alter the title.

(2) The authors state in the abstract that a phylogenetic-driven approach led to their identification of Gyc76C as a candidate receptor for ITPa. However, there are weaknesses in this claim. Firstly, because the hypothesis that Gyc76C may be involved in mediating effects of ITPa was first proposed ten years ago by Nagai et al. 2014, so this surely was the primary basis for investigating this protein. Nevertheless, investigating if there is correspondence in the phylogenetic distribution of ITP-type and Gyc76C-type genes/proteins is a valuable approach to addressing this issue. Unfortunately, the evidence presented is rather limited in scope. Essentially, the authors report that they only found ITP-type and Gyc76C-type genes/proteins in protostomes, but not in deuterostomes. What is needed is a more fine-grained analysis at the species level within the protostomes. Thus, are there protostome species in which both ITP-type and Gyc76C-type genes/proteins have been lost? Furthermore, are there any protostome species in which an ITP-type gene is present but an Gyc76C-type gene is absent, or vice versa? If there are protostome species in which an ITP-type gene is present but a Gyc76C-type gene is absent or vice versa, this would argue against Gyc76C being a receptor for ITPa. In this regard, it is noteworthy that in Figure 2A there are two ITP-type precursors in *C. elegans*, but there are no Gyc76Ctype proteins shown in the tree in Figure 2B. Thus, what is needed is a more detailed analysis of protostomes to investigate if there really is correspondence in the phylogenetic distribution of Gyc76C-type and ITP-type genes at the species level.

We thank the reviewer for this comment. While the previous study by Nagai et al had implicated Gyc76C in the ITP signaling pathway, how they narrowed down Gyc76C as a candidate was not reported. Therefore, our unbiased phylogenetic approach was necessary to ensure that we identified all suitable candidate receptors. Indeed, our phylogenetic analysis also identified Gyc32E as another candidate ITP receptor. However, we did not pursue this receptor further as our expression data (new Figure 4 Supplement 2) indicated that Gyc32E is not expressed in osmoregulatory tissues and therefore likely does not mediate the osmotic effects of ITPa.

We also appreciate the suggestion to perform a more detailed phylogenetic analysis for the peptide and receptor. We did not include *C. elegans* receptors in the phylogenetic analysis because they tend to be highly evolved and routinely cause long-branch attraction (see: Guerra and Zandawala 2024: https://doi.org/10.1093/gbe/evad108). We (specifically the senior author) have previously excluded *C. elegans* receptors in the phylogenetic analysis of GnRH and Corazonin receptors for similar reasons (see: Tian and Zandawala et al. 2016: 10.1038/srep28788).

Unfortunately, absence of a gene in a genome is hard to prove especially when they are not as high-quality as the genomes of model systems (e.g. Drosophila and mice). Moreover, given the concern of this reviewer that our physiological and behavioral data on ITPa and Gyc76C only provide correlative evidence, we decided against performing additional phylogenetic analysis which also provides correlative evidence. Our only goal with this analysis was to identify a candidate ITPa receptor. Since we have now functionally characterized this receptor using a heterologous system, we feel that the current phylogenetic analysis was able to successfully serve its purpose.

(3) The manuscript would benefit from a more comprehensive overview and discussion of published literature on Gyc76C in Drosophila, both as a basis for this study and for interpretation of the findings of this study.

We thank the reviewer for this comment. We have now included a broader discussion of Gyc76C based on published literature.

**Reviewer #3 (Public Review):**
Summary:The goal of this paper is to characterize an anti-diuretic signaling system in insects using *Drosophila melanogaster* as a model. Specifically, the authors wished to characterize a role of ion transport peptide (ITP) and its isoforms in regulating diverse aspects of physiology and metabolism. The authors combined genetic and comparative genomic approaches with classical physiological techniques and biochemical assays to provide a comprehensive analysis of ITP and its role in regulating fluid balance and metabolic homeostasis in Drosophila. The authors further characterized a previously unrecognized role for Gyc76C as a receptor for ITPa, an amidated isoform of ITP, and in mediating the effects of ITPa on fluid balance and metabolism. The evidence presented in favor of this model is very strong as it combines multiple approaches and employs ideal controls. Taken together, these findings represent an important contribution to the field of insect neuropeptides and neurohormones and have strong relevance for other animals.

We thank this reviewer for the positive and thorough assessment of our manuscript.

Strengths:Many approaches are used to support their model. Experiments were wellcontrolled, used appropriate statistical analyses, and were interpreted properly and without exaggeration.Weaknesses:No major weaknesses were identified by this reviewer. More evidence to support their model would be gained by using a loss-of-function approach with ITPa, and by providing more direct evidence that Gyc76C is the receptor that mediates the effects of ITPa on fat metabolism. However, these weaknesses do not detract from the overall quality of the evidence presented in this manuscript, which is very strong.

We agree with this reviewer regarding the need to provide additional evidence using a loss-of-function approach with ITPa. We now characterize the phenotypes following knockdown of ITP in ITP-producing cells (new Figure 9). Our results are in agreement with phenotypes observed following Gyc76C knockdown, lending further support that ITPa mediates its effects via Gyc76C. Unfortunately, we are not able to provide evidence that ITPa acts on Gyc76C in the fat body using the assay suggested by this reviewer (explained in detail below). Instead, we now provide direct evidence of Gyc76C activation by ITPa in a heterologous system (new Figure 7 and Figure 7 Supplement 1).

**Reviewer #1 (Recommendations For The Authors):**
Here, I have several extra concerns about the work as below:(1) The authors confirmed the function of ITPa in regulating both osmotic and metabolic homeostasis by specifically overexpressing ITPa driven by ITP-RCGal4 in adult flies (Figures. 5 and 7). Have authors ever tried to knock down ITP in ITP-RC-Gal4 neurons? What was the phenotype? Especially regarding the impact on metabolic homeostasis, does knocking down ITP in ITP neurons mimic the phenotypes of Gyc76C fat body knockdown flies?

We thank the reviewer for this suggestion. We now characterize the phenotypes following knockdown of ITP using ITP-RC-Gal4 (new Figure 9). Our results are in agreement with phenotypes observed following Gyc76C knockdown, lending further support that ITPa mediates its effects via Gyc76C.

The authors mentioned that the existing ITP RNAi lines target all three isoforms. It would be interesting if the authors could overexpress ITPa in ITPRC-Gal4>ITP-RNAi flies and confirm whether any phenotypes induced by ITP knockdown could be rescued. It will further confirm the role of ITPa in homeostasis regulation.

We thank the reviewer for this suggestion. Unfortunately, this experiment is not straightforward because knockdown with ITP RNAi does not completely abolish ITP expression (see Figure 9A). Hence, the rescue experiment needs to be ideally performed in an ITP mutant background. However, ITP mutation leads to developmental lethality (unpublished observation) so we cannot generate all the flies necessary for this experiment. Therefore, we cannot perform the rescue experiments at this time. In future studies, we hope to perform knockdown of specific ITP isoforms using the transgenes generated here (Xu et al 2023: 10.1038/s41586-023-06833-8).

(2) In Figures 5A and B, the authors nicely show the increased release of ITPa under desiccation by quantifying the ITPa immunolabelling intensity in different neuronal populations. It may be induced by the increased neuronal activity of ITPa neurons under the desiccated condition. Have the authors confirmed whether the activity of ITPa-expressing neurons is impacted by desiccation?The TRIC system may be able to detect the different activity of those neurons before and after desiccation. This may further explain the reduced ITPa peptide levels during desiccation.

We thank the reviewer for this suggestion. We have now monitored the activity of ITPa-expressing neurons using the CaLexA system (Masuyama et al 2012: 10.3109/01677063.2011.642910). Our results indicate that ITPa neurons are indeed active under desiccation (new Figure 8A and B). These results are also in agreement with ITPa immunolabelling showing increased peptide release during desiccation (new Figure 8C and D). Together, these results show that ITPa neurons are activated and release ITPa under desiccation.

(3) What about the intensity of ITPa immunolabelling in other ITPa-positive neurons (e.g., VNC) under desiccation? If there is no change in other ITPa neurons, it will be a good control.

We thank the reviewer for this suggestion. Unfortunately, ITPa immunostaining in VNC neurons is extremely weak preventing accurate quantification of ITPa levels under different conditions. We did hypothesize that ITPa immunolabelling in clock neurons (5^th^-LN_v_ and LN_d_^ITP^) would not change depending on the osmotic state of the animal. However, our results (Figure 8C and D) indicate that ITPa from these neurons is also released under desiccation. Interestingly, LNd^ITP^, which also coexpress Neuropeptide F (NPF) have recently been implicated in water seeking during thirst (Ramirez et al, 2025: 10.1101/2025.07.03.662850). Our new connectomic-driven analysis shows that these neurons can receive thermo/hygrosensory inputs (new Figure 13E). Hence, it is conceivable that other ITPa-expressing neurons also release ITPa during thirst/desiccation.

(4) The adult stage, specifically overexpression of ITPa in ITP neurons, does show significant phenotypes compared to controls in both osmotic and metabolic homeostasis-related assays. It would be helpful if authors could show how much ITPa mRNA levels are increased in the fly heads with ITPa overexpression (under desiccation & starvation or not).

We thank the reviewer for this suggestion. We have now included immunohistochemical evidence showing increase in ITPa peptide levels in flies with ITPa overexpression (new Figure 10A). We feel that this is a better indicator of ITPa signaling level instead of ITPa mRNA levels.

(5) Another question concerns the bloated abdomens of ITPa-overexpressing flies. Are the bloated abdomens of ITPa OE female flies (Figure 5E) due to increased ovary size (Figure 7G)? Have the authors also detected similar bloated abdomens in male flies with ITPa overexpression? Since both male and female flies show more release of ITPa during the desiccation.

We thank the reviewer for this comment. The bloated abdomen phenotype seen in females can be attributed to increased water content since we see a similar phenotype in males (see Author response image 1 below).

**Author response image 1. sa4fig1:** 

**Reviewer #2 (Recommendations For The Authors):**
(1) Page 1 - change "Homeostasis is obtained by" to "Homeostasis is achieved by".

Changed

(2) Page 1 - change "Physiological responses" to "Physiological processes".

Changed

(3) Page 2 - Change "Recently, ITPL2 was also shown to mediate anti-diuretic effects via the tachykinin receptor" to "Recently, ITPL2 was also shown to exert anti-diuretic effects via the tachykinin receptor".

Changed

(4) Page 9 - "(C) Adult-specific overexpression of ITPa using ITP- RC-GAL4TS (ITP-RC-T2A-GAL4 combined with temperature-sensitive tubulinGAL80) increases desiccation" Unless I am misunderstanding Fig 5C, I think what is shown is that overexpression of ITPa prolongs survival during a period of desiccation. I am not sure what the authors mean by "increases desiccation". In the text (page 9) the authors state "ITPa overexpression improves desiccation tolerance, which is a much clearer statement than what is in the figure legend.

We thank the reviewer for identifying this oversight. We have now changed the caption to “increases desiccation tolerance”.

(5) Page 11 - The authors conclude that "increased ITPa signaling results in phenotypes that largely mirror those seen following Gyc76C knockdown in the fat body, providing further support that ITPa mediates its effects via Gyc76C." Use of the term "largely mirror" seems inappropriate here because there are opposing effects- e.g. decreased starvation resistance in Figure 6A versus increased starvation resistance in Figure 7A.Perhaps there is a misunderstanding of what is meant by "mirroring" - it means the same, not the opposite.

We thank the reviewer for this comment. We agree that the use of the term “largely mirrors” to describe the effects of ITPa overexpression and Gyc76C knockdown is not appropriate and have changed this sentence as follows: “Taken together, the phenotypes seen following Gyc76C knockdown in the fat body largely mirror those seen following ITP knockdown in ITP-RC neurons, providing further support that ITPa mediates its effects via Gyc76C.”

(6) Page 12 - There appear to be words missing between "neurons during desiccation, as well as their downstream" and "the recently completed FlyWire adult brain connectome"

We thank the reviewer for highlighting this mistake. We have changed the sentence as following: “Having characterized the functions of ITP signaling to the renal tubules and the fat body, we wanted to identify the factors and mechanisms regulating the activity of ITP neurons during desiccation, as well as their downstream neuronal pathways. To address this, we took advantage of the recently completed FlyWire adult brain connectome (Dorkenwald et al., 2024, Schlegel et al., 2024) to identify pre- and post-synaptic partners of ITP neurons.”

(7) Page 15 - "can release up to a staggering 8 neuropeptides" - I suggest that the word "staggering" is removed. The notion that individual neurons release many neuropeptides is now widely recognised (both in vertebrates and invertebrates) based on analysis of single-cell transcriptomic data.

Removed staggering.

(8) Page 16 - "(Farwa and Jean-Paul, 2024)" - this citation needs to be added to the reference list and I think it needs to be changed to "Sajadi and Paluzzi, 2024".

We thank the reviewer for highlighting this oversight. The correct citation has now been added.

(9) It is noteworthy that, based on a PubMed search, there are at least thirteen published papers that report on Gyc76C in Drosophila (PMIDs: 34988396, 32063902, 27642749, 26440503, 24284209, 23862019, 23213443, 21893139, 21350862, 16341244, 15485853, 15282266, 7706258). However, none of these papers are discussed/cited by the authors. This is surprising because the authors' hypothesis that Gyc76C acts as a receptor for ITPa surely needs to be evaluated and discussed with reference to all the published insights into the developmental/physiological roles of this protein.

We thank the reviewer for this comment. Some of the references mentioned above (21350862, 16341244, 15485853) mainly report on soluble guanylyl cyclases and not membrane guanylyl cyclase like Gyc76C. Based on other studies on Gyc76C and its role in immunity and development, we have now expanded the discussion on additional roles of ITPa.

**Reviewer #3 (Recommendations For The Authors):**
I have only a few comments that will help the authors strengthen a couple of aspects of their model.(1) The case for Gyc76C as a receptor for ITPa in regulating fluid homeostasis is clear, given the experiments the authors carried out where they applied ITPa to tubules and showed that the effects of ITPa on tubule secretion were blocked if Gyc76C was absent in tubules. This approach, or something similar, should be used to provide conclusive proof that ITPa's metabolic effects on the fat body go through Gyc76C.At present (unless I missed it) the authors only show that gain of ITPa has the opposite phenotype to fat body-specific loss of Gyc76C. While this would be the expected result if ITPa/Gyc76C is a ligand-receptor pair, it is not quite sufficient to conclusively demonstrate that Gyc76C is definitely the fat body receptor. Ex vivo experiments such as soaking the adult fat body carcasses with and without Gyc76C in ITPa and monitoring fat content via Nile Red could be one way to address this lack of direct evidence. The authors could also make text changes to explicitly mention this lack of conclusive evidence and suggest it as a future direction.

We thank the reviewer for this comment. We have now conclusively demonstrated that Gyc76C is activated by ITPa in a heterologous assay (new Figure 7 and Figure 7 Supplement 1). With this evidence, we can confidently claim that ITPa can mediate its actions via Gyc76C in various tissues including the Malpighian tubules and fat body. Nonetheless, we liked the suggestion by this reviewer to perform the ex vivo assay and test the effect of ITPa on the fat body. Unfortunately, it is challenging to do this because increased ITPa signaling (chronically using ITPa overexpression) results in increased lipid accumulation in the fat body in vivo. Therefore, we would likely not see the effect of ITPa addition in an ex vivo fat body preparation since lipogenesis will not occur in the absence of glucose. However, ITPa could counteract the effects of other lipolytic factors such as adipokinetic hormone (AKH). To test this hypothesis, we monitored fat content in the fat body incubated with and without AKH (see Author response image 2 below showing representative images from this experiment). Since we did not observe any differences in fat levels between these two conditions, we were unable to test the effects of ITPa on AKH-activity using this assay.

**Author response image 2. sa4fig2:** 

(2) I did not see any loss of function data for ITPa - is this possible? If so this would strengthen the case for a 1:1 relationship between loss of ligand and loss of receptor. Alternatively, the authors could suggest this as an important future direction.

We agree with this reviewer regarding the need to provide additional evidence using a loss-of-function approach with ITPa. We have now characterized the phenotypes following knockdown of ITP in ITP-producing cells (new Figure 9). Our results are in agreement with phenotypes observed following Gyc76C knockdown, lending further support that ITPa mediates its effects via Gyc76C.

(3) For clarity, please include the sex of all animals in the figure legend. Even though the methods say 'females used unless otherwise indicated' it is still better for the reader to know within the figure legend what sex is displayed.

We thank the reviewer for this suggestion and have now included sex of the animals in the figure legends.

(4) Please state whether females are mated or not, as this is relevant for taste preferences and food intake.

We apologize for this oversight. We used mated females for all experiments. This has now been included in the methods.

(5) More discussion on the previous study on metabolic effects of ITP in this study compared with past studies would help readers appreciate any similarities and/or differences between this study and past work (Galikova 2018, 2022)

We thank the reviewer for this suggestion. Unfortunately, it is difficult to directly compare our phenotypes with the metabolic effects of ITP reported in Galikova and Klepsatel 2022 because the previous study used a ubiquitous driver (Da-GAL4) to manipulate ITP levels. Ectopically overexpressing ITPa in non-ITP producing cells can result in non-physiological phenotypes. This is evident in their metabolic measurements where both global overexpression and knockdown of ITP results in reduced glycogen and fat levels, and starvation tolerance. Moreover, ITP-RC-GAL4 used in our study to overexpress and knockdown ITPa is more specific than the Da-GAL4 used previously. Da-GAL4 would include other ITP cells (e.g. ITP-RD producing cells). Since ITP is broadly expressed across the animal, it is difficult to parse out the phenotypes of ITPa and other isoforms using manipulations performed with Da-GAL4. We have mentioned this limitation in the results for ITP knockdown as follows: “A previous study employing ubiquitous ITP knockdown and overexpression suggests that Drosophila ITP also regulates feeding and metabolic homeostasis (Galikova and Klepsatel, 2022) in addition to osmotic homeostais (Galikova et al., 2018). However, given the nature of the genetic manipulations (ectopic ITPa overexpression and knockdown of ITP in all tissues) utilized in those studies, it is difficult to parse the effects of ITP signaling from ITPa-producing neurons.”